# KLAS: Using Similarity to Stitch Neural Networks for Improved Accuracy-Efficiency Tradeoffs

**Debopam Sanyal**[1], **Anantharaman Iyer**[1], **Alind Khare**[2], **Trisha Jain**[1], **Akshay Jajoo**[3],
**Myungjin Lee**[3], **Clayton Kerce**[4], **Alexey Tumanov**[1]
[1]Georgia Institute of Technology, [2]Microsoft M365 Research, [3]Cisco Research,
[4]Georgia Tech Research Institute
`debopam.sanyal@gatech.edu`

## Abstract

Given the wide range of deployment targets, flexible model selection is essential for optimizing performance within a given compute budget. Recent work demonstrates that stitching pretrained models within a model family enables cost-effective interpolation of the accuracy-efficiency tradeoff space. Stitching transforms intermediate activations from one pretrained model into another, producing a new interpolated stitched network. Such networks provide a pool of deployment options along the accuracy-efficiency spectrum. However, existing stitching approaches often yield suboptimal tradeoffs and lack generalizability, as they primarily rely on heuristics to select stitch configurations. We argue that constructing improved accuracy-efficiency tradeoffs requires explicitly capturing and leveraging the *similarity* between pretrained models being stitched. To this end, we introduce **KLAS**, a novel stitch selection framework that automates and generalizes stitch selection across model families by leveraging KL divergence between intermediate representations. KLAS identifies the most promising binary stitches from the $\mathcal{O}(k^2 n^2)$ possibilities for $k$ pretrained models of depth $n$. Through comprehensive experiments, we demonstrate that KLAS improves the accuracy-efficiency curve of stitched models at the same finetuning cost as baselines. KLAS achieves up to $1.21\%$ higher ImageNet-1K top-1 accuracy at the same computational cost, or maintains accuracy with a $1.33\times$ reduction in FLOPs.

## 1 Introduction

Large pretrained vision transformers (ViTs) and convolutional neural networks (CNNs) have transformed computer vision Devlin et al. (2019); Brown et al. (2020); Peters et al. (2018); Srivastava et al. (2025), powering applications from self-driving cars Ouyang et al. (2019) to security systems Chen et al. (2021); Ji et al. (2021) and social media platforms Bai et al. (2018). Trained on vast datasets and distributed via repositories such as HuggingFace Wolf (2019), PyTorchHub Paszke et al. (2019), and TensorFlow Model Zoo Abadi et al. (2016), these models achieve strong performance across tasks including image classification, object detection, semantic segmentation, and face recognition Liu et al. (2022); Ravi et al. (2024); An et al. (2022). Yet, deployment environments vary dramatically from powerful cloud servers Hazelwood et al. (2018) to resource-constrained edge devices He et al. (2018); David et al. (2021). This poses a fundamental challenge: *how can we efficiently deploy pretrained models under diverse computational budgets without sacrificing accuracy?*

Neural architecture search (NAS) Cai et al. (2020); Zoph & Le (2016) aims to address this challenge by discovering architectures that balance accuracy and efficiency, but conventional one-to-one NAS requires prohibitively high GPU cost. One-to-many paradigms, such as one-shot NAS Liu et al. (2018); Guo et al. (2020) mitigate this burden by training a single supernetwork, while zero-shot NAS Li et al. (2023); Lin et al. (2021) avoids costly training altogether by ranking candidate architectures using gradient-based proxies. Despite these advances, both approaches remain limited: they are confined to a single model design space and cannot exploit the rich pool of pretrained models already available. This limitation prevents one-to-many NAS from satisfying the three key requirements of effective architecture search: (1) strong accuracy-efficiency tradeoffs, (2) generalizability across diverse model architectures, and (3) low computational cost.

Many-to-many NAS with model stitching Pan et al. (2023); Bansal et al. (2021); Yang et al. (2022b); Pan et al. (2024) offers a promising solution by combining blocks from different pretrained "anchor" models to form stitched networks that interpolate accuracy-efficiency tradeoffs. For example, SN-

Net Pan et al. (2023), finetunes lightweight stitching layers (simple linear transformations) between anchors to drastically reduce cost. A key challenge, however, lies in selecting stitch configurations (i.e., choosing which anchors and blocks to stitch). Even with only two anchor models of depth $n$, there are $\mathcal{O}(n^2)$ possible stitch configurations, making exhaustive search computationally infeasible. Yet, SN-Net employs naive heuristic-based selection (e.g., stitching adjacent anchors and blocks), which limits both the accuracy-efficiency tradeoff and generalizability across model families.

We propose a novel framework, **KLAS (KL divergence based Anchor Stitching)**, which replaces heuristic-based stitching with a principled similarity-driven approach. As illustrated in Fig. 1, KLAS automatically selects anchors and block pairs by evaluating the compatibility of their activations using Kullback-Leibler (KL) divergence Cover (1999). Unlike alternatives such as mean square error Jadon et al. (2024), cross-entropy Mao et al. (2023), or CKA Kornblith et al. (2019), KL divergence effectively captures distributional differences between decision boundaries, enabling more accurate and efficient stitching with minimal finetuning. In doing so, KLAS addresses two key stitching questions: (1) *can network B produce similar outputs when driven by transformed internal representations from network A*? and (2) *can such transformations be effective with minimal finetuning*?

We evaluate KLAS on ImageNet-1K Deng et al. (2009) and CIFAR-100 Krizhevsky et al. (2009) across multiple ViT-based and CNN-based model families and show that it consistently improves the accuracy-efficiency tradeoff over SN-Net. KLAS achieves up to $1.21\%$ higher top-1 accuracy at the same FLOPs, or alternatively, a $1.33\times$ reduction in FLOPs at equivalent accuracy. We further show benefits of KLAS on large language models (LLMs) using TruthfulQA Lin et al. (2022). Our key contributions are: (1) identifying the limitations of heuristic-based many-to-many NAS approaches such as SN-Net, (2) introducing KL divergence as a principled similarity

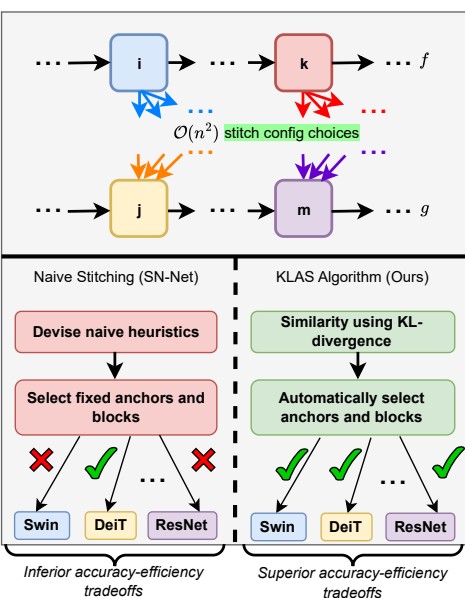

Figure 1: Comparison of stitching strategies. Existing many-to-many NAS approaches such as SN-Net rely on heuristic-based stitch selection, fixing anchors and blocks and yielding suboptimal accuracy-efficiency tradeoffs. In contrast, our proposed KLAS framework leverages KL divergence to measure similarity between intermediate representations and automatically identify promising stitch configurations, producing superior accuracy-efficiency tradeoffs across diverse model families.

metric for selecting stitch configurations without instantiating or training a stitch layer, (3) presenting the KLAS framework for efficient, generalizable, and low-cost stitching, and (4) demonstrating consistent improvements over SN-Net across model families and datasets.

## 2 RELATED WORK

**Model Stitching:** Model stitching was originally introduced as a tool for analyzing representation equivalence across neural networks. Lenc & Vedaldi (2015) showed that early layers of one network could be connected to later layers of another via simple linear transformations, revealing functional similarities between disparate architectures. Building on this, Bansal et al. (2021) demonstrated that stitching remains effective even across models with different architectures or training protocols. While these studies established stitching as a powerful analytical tool, they did not investigate its potential for practical deployment optimization.

Recent efforts have begun bridging this gap. SN-Net Pan et al. (2023) operationalized stitching for neural architecture search by combining pretrained anchor models from the same family, reducing training costs by $22\times$ compared to traditional NAS. However, SN-Net relies on heuristic stitching strategies, such as connecting adjacent anchors ("nearest stitching") or assuming block similarity ("paired / unpaired stitching"), that ignore actual compatibility between blocks. ESTA He et al. (2024) extended SN-Net to large language models by applying the same heuristic, layer-count-based stitching strategy when combining pretrained LLM blocks. StitchLLM Hu et al. (2025) further adopts this identical heuristic within a systems framework focused on resource-efficient serving, but does not

propose a new stitch selection criterion. Concurrently, Csiszárik et al. (2021) systematically analyzed stitching performance across block combinations, finding no correlation between common similarity metrics (e.g., CKA) and stitched model accuracy. Similarly, Balogh & Jelasity (2025) showed that direct similarity measures often fail to reflect task performance, advocating task loss matching as a more reliable criterion. Together, these findings underscore the inadequacy of existing approaches for automatically identifying effective stitch points, a challenge that KLAS addresses through activation distribution analysis using KL divergence.

**Neural Architecture Search:** Traditional NAS methods Cai et al. (2020); Sahni et al. (2021); Pham et al. (2018); Zoph & Le (2016); Wang et al. (2021) incur prohibitive computational costs, often requiring thousands of GPU hours to evaluate candidate architectures. One-shot NAS Liu et al. (2018) mitigates this burden by training a single supernetwork with shared weights across subnetworks, but still demands substantial resources for supernetwork training and remains restricted to a single architectural family. Zero-shot NAS Lin et al. (2021) further reduces cost by predicting subnetwork performance from architectural indicators, yet the final models still require full training from scratch. **Cascaded inference** has long been explored as a means to reduce compute by exiting early when predictions are confident Viola & Jones (2001). Approaches include big / little-style two-stage models Park et al. (2015), multi-exit deep cascades with learned entropy thresholds Wang et al. (2017); Guan et al. (2017); Bolukbasi et al. (2017), and exit-by-confidence ensemble routing Streeter (2018); Inoue (2019). Recent work Wang et al. (2020); Lebovitz et al. (2023) extends cascades to four-way model ensembles using softmax-based confidence and input resolution scaling.

Stitching circumvents these limitations by leveraging pretrained model zoos rather than training networks from the ground up. Unlike DeRy Yang et al. (2022a), which reassembles pretrained components for individual resource constraints, KLAS systematically analyzes activation distributions to generate a continuous Pareto front of deployable models. This approach preserves cost efficiency by requiring only stitch-layer finetuning, while avoiding architectural constraints by enabling seamless integration of transformers, convnets, and hybrid models.

**Transformers and ConvNets:** The rise of ViTs Dosovitskiy et al. (2020) has introduced new challenges for efficient deployment. Prior efforts optimized individual ViTs through token pruning Pan et al. (2021), quantization Li et al. (2022), or dynamic inference Rao et al. (2021), but these methods remain architecture-specific and require retraining under different resource constraints. Cross-architectural stitching approaches such as SN-Net Pan et al. (2023) explored combining ViTs, yet were limited to same-family stitching due to reliance on weak heuristics.

KLAS breaks this barrier by introducing KL divergence between block activations as an intuitive similarity measure, enabling stitching across diverse architectural families. Whereas prior work Kornblith et al. (2019) found standard similarity metrics ineffective for predicting stitching performance, KLAS directly measures how well the outputs of one block can serve as the inputs to another, regardless of architecture. This capability allows KLAS to produce viable hybrids spanning hierarchical Swin Transformers and residual CNNs.

## 3 BACKGROUND AND MOTIVATION

### 3.1 NOTATIONS FOR MODEL STITCHING

Let $f : \mathcal{X} \to \mathcal{Y}$ be a feedforward neural network with $m$ blocks (i.e., layers): $f = f_m \circ \cdots \circ f_1$ where $f_i : A_{i-1}^f \to A_i^f$ maps the activation space of block $i-1$ to that of block $i$. By definition, $A_0^f = X$ (i.e., the input samples). For model stitching, we introduce the notations $f_{\leq i} = f_i \circ \cdots \circ f_1$ and $f_{>i} = f_m \circ \cdots \circ f_{i+1}$. Given two frozen networks $f$ and $g$, and one block from each network, $i \in f$ and $j \in g$, the goal of stitching is to find out if $g_{>j}$, which we will refer to as the *target*, can achieve its function using the representation of $f_{\leq i}$, which we call the *source*. In examining this, we attempt to find a transformation layer $T : A_i^f \to A_j^g$ such that the stitched network $g_{>j} \circ T \circ f_{\leq i}$ is functionally similar to $g$. Here, we term $(i, j)$ as a stitch configuration, meaning that the $i$-th layer of $f$ is stitched to the $j$-th layer of $g$. Throughout this paper, we refer to $T$ as the stitching layer. For an input $x \in \mathcal{X}$, we denote $f_{\leq i}(x)$ as its source representation and $g_{\leq j}(x)$ as its target representation. Transformer blocks with the same hidden dimensions are often grouped into *stages*. We omit the stage index in our notations for simplicity as stitching only takes place between same stage blocks.

## 3.2 SIMILARITY OF NEURAL NETWORKS

A central theme in analyzing neural networks is understanding the representations learned by internal layers, since these insights are crucial for both interpreting and improving deep learning systems. A fundamental question in this context is: *when are two representations similar*? Prior work Bansal et al. (2021); Kornblith et al. (2019); Yosinski et al. (2014) extends this inquiry by asking a complementary question: *in what way are two representations similar, once similarity is established*? Together, these perspectives give rise to two notions of similarity: (1) *representational similarity*, which characterizes structural relationships between learned features, and (2) *functional similarity*, which evaluates their interchangeability in downstream tasks. We explain below why both are essential.

**Representational similarity** captures the alignment of geometric structures between intermediate activations, enabling the stitching layer ($T$) to learn a simple mapping (e.g., affine transformation) from the source to the target. For instance, convolutional layers with comparable channel counts and spatial dimensions often permit lightweight $1 \times 1$ convolutions as effective stitching layers. Conversely, mismatched representations may require complex transformations, which can degrade performance or even cause the stitched network to fail. A widely used metric for representational similarity is *Centered Kernel Alignment* (CKA) Kornblith et al. (2019), which leverages kernel functions to embed data into a higher-dimensional space where linear relationships between representations become more discernible.

**Functional similarity** evaluates whether the stitched network preserves the target's original output behavior, typically measured via task accuracy. High functional similarity implies that the source representations contain sufficient task-relevant information for the target, even if their geometric structures differ. Two common approaches include *task loss matching* (TLM), which optimizes the stitching layer by minimizing downstream task loss (e.g., mean squared error (MSE) or cross-entropy (CE)), and direct matching (DM), which aligns source activations after the stitch layer and target activations through linear regression Balogh & Jelasity (2025).

## 3.3 SHORTCOMINGS OF CURRENT SIMILARITY MEASURES

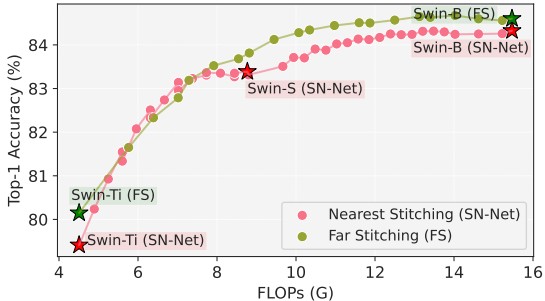

| Metric | Type | Overlap |
|---|---|---|
| CKA | ✗ (Unsupervised) | 5.5% |
| MSE | ✓ (Supervised) | 27.8% |
| CE | ✓ (Supervised) | 22.2% |
| DM | ✓ (Supervised) | 33.3% |
| SN-Net | ✗ (Unsupervised) | 61.1% |
| *KL-Div(Ours)* | ✓ (Supervised) | **88.9%** |

Table 1: Effectiveness of similarity metrics for recovering Swin Ti-B stitching, measured by overlap with the Ti-B stitch configurations. Existing metrics (CKA, MSE, CE, DM) and SN-Net heuristics achieve at most 61.1% overlap, while our KL divergence based approach achieves 88.9%, demonstrating its reliability as a similarity measure.

Figure 2: Comparison of nearest stitching (SN-Net) and far stitching (FS) on the Swin family. Far stitching between Ti and B models (green curve) achieves a superior accuracy-efficiency tradeoff, surpassing nearest stitching (pink curve) by up to 0.9% accuracy at equivalent FLOPs. Stars mark pretrained anchors (Ti, S, B).

In our empirical study of similarity metrics for model stitching, we identify a key limitation of the naive heuristics adopted by SN-Net. Specifically, SN-Net assumes that combining *nearest stitching* for anchors with *paired / unpaired stitching* for blocks yields optimal results across model families. Nearest stitching restricts anchor connections to models of comparable complexity or performance, such as the Ti-S-B stitching style in Swin transformers. However, our experiments reveal that direct *far stitching* between Ti and B models (i.e., Ti-B stitching) achieves a superior accuracy-efficiency tradeoff, improving accuracy by up to 0.9% at equivalent FLOPs, as shown in Fig. 2. For this study, we apply paired / unpaired block stitching with Swin-Ti/S/B models pretrained on ImageNet-22K, followed by finetuning the stitched supernetwork for 50 epochs on ImageNet-1K.

We further examine whether existing similarity measures can automatically recover the Swin Ti-B stitching by evaluating the percentage overlap of stitch configurations. As shown in Tab. 1, commonly

used metrics such as MSE, CE, CKA, and DM fail to do so, while KL divergence is successful. The gap is particularly pronounced for CKA, which recovers only $5.5\%$ of the configurations. We provide explanations in §4.1 and further studies in §5.1 and App. D.

# 4 PROPOSED APPROACH

We present KLAS, a generalizable stitch selection technique that identifies promising configurations for favorable accuracy-efficient tradeoffs, at no additional finetuning cost over baselines. Central to KLAS is the observation that stitching success hinges on activation similarity, which determines whether two networks can be integrated through a linear transformation layer. When stitching the source network's early layers ($f_{\leq i}$) to the target network's later layers ($g_{>j}$), the stitching layer ($T$) must map the source activations ($\mathcal{A}_i^f$) to the target activations ($\mathcal{A}_j^g$) *while keeping finetuning costs low and preserving task performance*. The first requirement corresponds to representational similarity, and the second to functional similarity. At its core, KLAS employs KL divergence, which we show in §4.1 satisfies both objectives, unlike prior metrics that capture only one. This dual alignment ensures stitching viability: *representational alignment simplifies stitch-layer learning, while functional alignment guarantees good downstream task performance*.

## 4.1 KL DIVERGENCE AS A SIMILARITY METRIC

KL divergence satisfies the dual objectives of measuring both representational and functional similarity by directly quantifying alignment between activation distributions and their downstream task performance. As a representational similarity metric, KL divergence captures distributional differences between intermediate activations, indicating whether two blocks generate patterns that can be mapped via lightweight transformations. At the same time, it serves as a functional similarity metric by assessing how well one block's outputs preserve the target model's decision boundaries: *a low KL value* implies that the source block provides task-relevant information that the target can leverage with minimal finetuning. This dual capability arises from KL divergence's grounding in information theory, as it evaluates both statistical distance between representations (representational alignment) and the retention of class-discriminative information (functional alignment). By minimizing KL divergence between intermediate activations, we ensure that the stitching layer requires only minor adjustments to align representations while preserving the target's original performance: an advantage over metrics that emphasize either representational alignment or task loss matching.

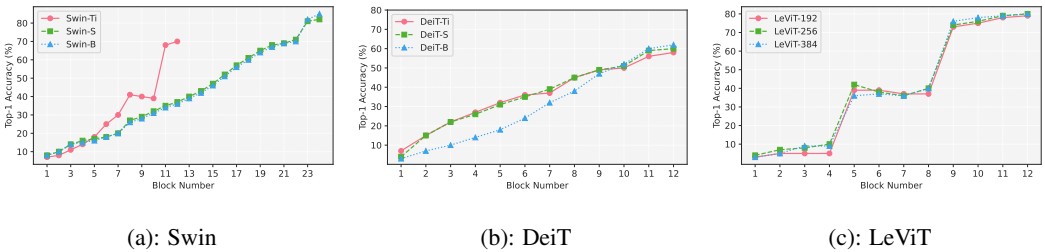

(a): Swin  (b): DeiT  (c): LeViT

Figure 3: Probe training convergence across architectures. Top-1 accuracy of linear probes trained for four epochs is shown for each block in (a) Swin, (b) DeiT, and (c) LeViT families. Probes converge rapidly, demonstrating that meaningful intermediate representations can be extracted with negligible, one-time probe training cost.

**Linear Probe Classifiers:** Formally, KL divergence measures the statistical distance from a probability distribution $p$ to a target distribution $q$, denoted as $D_{KL}(p||q)$. We use linear classifier probes Alain & Bengio (2018), placed after different blocks, to estimate the KL divergence between the softmax probability distributions of activations. Each block in an anchor model is equipped with a simple $1 \times 1$ convolutional probe, for instance, Swin-B requires 24 probes, while DeiT-S requires 12. Training all probes independently would be very expensive. Instead, we introduce a unified architecture, *ProbeNet*, which jointly trains all probes by activating one at a time within a single forward-backward pass. This design adds only a *negligible one-time cost*, as probes converge rapidly in our setup (by epoch 4; Fig. 3). For example, training all 24 probes in Swin-B required just $0.25$ GPU days in total, rather than $24 \times 0.25$ GPU days if trained separately. The pseudocode for ProbeNet is provided in Alg. 1 in App. A. By avoiding redundant inference and backpropagation, ProbeNet makes large-scale linear probing both efficient and practical.

Let $P_i^f(x)$ denote the softmax probability distribution produced by the probe inserted after block $i$ of anchor model $f$ on input $x \in \mathcal{D}_v$, where $\mathcal{D}_v$ is the validation split. Similarly, let $P_j^g(x)$ represent the distribution from the probe after block $j$ of anchor model $g$. After training the linear probes on the training split ($\mathcal{D}_t$), we compute the similarity score $\Theta$ as defined in Eq. 1. We outline the procedure for obtaining KL divergence based similarity scores for all source-target block pairs using classifier probes in Alg. 2 in App. A. In this work, we fix the direction of stitching such that the *source anchor always has lower complexity than the target anchor*. Experiments in Pan et al. (2023) indicate that this choice yields more stable training and consistently better performance compared to stitching in the reverse direction.

$$\Theta(P_i^f, P_j^g) = \frac{\sum_{x \in \mathcal{D}_v} D_{KL}\big(P_i^f(x) \,||\, P_j^g(x)\big)}{|\mathcal{D}_v|}; \qquad \Gamma(i,j) = \frac{\overbrace{\Theta(P_i^f, P_j^g)}^{\text{cross-anchor activation distance } (\Omega)}}{\underbrace{\Theta(P_j^g, P_{j+1}^g)}_{\text{intra-anchor block capacity } (\Sigma)}} \qquad (2)$$

$$(1)$$

## 4.2 THE KLAS FRAMEWORK: KL DIVERGENCE BASED ANCHOR STITCHING

**Anchor Selection using Last Block KL Divergence:** The first step in stitching is to identify suitable anchors (i.e., pretrained models). We achieve this by computing the KL divergence between the softmax probability distributions produced by the last block of each anchor. A low KL divergence indicates that two models have similar decision boundaries and class-confidence distributions, making them more compatible for stitching. This provides a principled anchor selection strategy, in contrast to the heuristic *nearest stitching* used in SN-Net.

**Block Selection using KL Divergence:** Once the anchors are chosen, we select stitchable block pairs using a stitch score $\Gamma$, defined in Eq. 2 and demonstrated in Fig. 4. Unlike SN-Net's naive *paired / unpaired stitching* heuristic for block selection, our method evaluates the compatibility of block pairs without instantiating or training a stitch layer. This design is central to KLAS's efficiency: *candidate stitched models are assessed without incurring any finetuning cost*. When stitching a source block $i$ from anchor $f$ to a target block $j$ in anchor $g$, the resulting stitched model $g_{>j} \circ T \circ f_{\leq i}$ interpolates between $f$ and $g$ in both accuracy and FLOPs. Thus, the target anchor's accuracy serves as an upper bound for any stitched network derived from it, i.e., $\text{Acc}(g)$ is the maximum achievable accuracy of $g_{>j} \circ T \circ f_{\leq i}, \forall i, j$.

(1) **Cross-anchor activation distance** [$\Omega = \Theta(P_i^f, P_j^g)$] quantifies the transformation a *hypothetical* stitch layer ($T$) would need to apply to the source activations ($A_i^f$) in order to match the target activations ($A_j^g$). In other words, it measures how much the stitched model $g_{>j} \circ T \circ f_{\leq i}$ would have to modify its intermediate representations to approximate the behavior of the target model $g$. A smaller $\Omega$ implies that the activations after layer $i$ in $f$ are already close to the activations after layer $j$ in $g$, making the block pair $(i, j)$ more favorable for stitching. It is important to emphasize that this is a purely *hypothetical* measure: since the models are not stitched yet, the $(j+1)$-th block in $g$ does not actually process activations from $f$. Rather, $\Omega$ reflects the expected burden on the stitch layer alone if such a connection were to be instantiated.

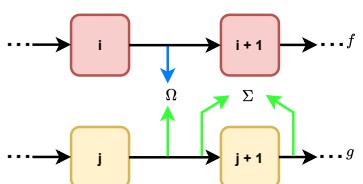

Figure 4: $\Omega$ measures how closely the activations of source block $i$ from $f$ align with those of target block $j$ in $g$, reflecting the transformation burden on the *hypothetical* stitch layer. $\Sigma$ measures how much block $j+1$ in the target anchor $g$ transforms its inputs relative to block $j$, indicating block $(j+1)$'s ability to absorb mismatched representations from the source.

(2) **Intra-anchor block capacity** [$\Sigma = \Theta(P_j^g, P_{j+1}^g)$] captures how much a block in a pretrained anchor transforms its input representations, measured by comparing the outputs of consecutive blocks within the same anchor. A large $\Sigma$ indicates a substantial shift in activations from block $j$ to block $j+1$ in model $g$. Since $g$ achieves any stitched model's upper-bound accuracy, this implies that block $j+1$ has *high learning capacity*, i.e., it plays a critical role in refining representations for task performance. High-capacity blocks are especially valuable when constructing stitched models, as they can absorb distributional differences from the source while maintaining task-relevant transformations. Thus, incorporating such blocks increases the likelihood of achieving strong downstream accuracy without a high training cost.

**KLAS algorithm formulation:** We employ probe classifiers to project the activation spaces of intermediate blocks onto the output space $\mathcal{Y}$. Since stitching is restricted to occur within the same stage, we omit explicit stage indices for notational simplicity (see §3.1). During the stitch finetuning phase, we select a set of candidate stitch configurations ($S$) using a tunable threshold ($\tau$). These candidates are drawn from a set of FLOPs buckets ($\mathcal{B}$), which partition the computational space between $f$ and $g$. The bucket granularity controls the density of the resulting accuracy-efficiency tradeoff curve. To ensure coverage, we choose $\mathcal{B}$ such that each block in the target anchor $g$ has at least one stitch configuration represented in $S$. Formally, the candidate set $S$ is defined as:

$$\mathcal{S} = \bigcup_{b \in \mathcal{B}} \mathcal{R}_b^*; \; \mathcal{R}_b^* = \left( \left\{ (i^*, j^*) \mid \Gamma(i,j) \leq \tau, \; \forall (i,j) \in b \right\} \bigcup \left\{ (i^*, j^*) = \underset{(i,j) \in b}{\arg \min} \, \Gamma(i,j) \right\} \right) \tag{3}$$

where $\Gamma(i,j)$ denotes the stitch score for block pair $(i,j)$. This formulation in Eq. 3 balances computational efficiency with stitch quality by ensuring both bucket-level coverage and threshold-based filtering. The complete KLAS algorithm is presented in Alg. 3 in App. A.

## 5 EXPERIMENTS AND EVALUATION

Please refer to App. B for details on the experimental setup. All results in this section are reported after finetuning the stitched network for *50 epochs* with identical settings following SN-Net Pan et al. (2023). The end-to-end overhead of KLAS (ProbeNet + Stitched network finetuning) on $8 \times$ A40 NVIDIA GPUs is *16 hours* for the Swin model family.

### 5.1 MEASURING SIMILARITY USING KL DIVERGENCE

Tab. 2 shows that KLAS significantly outperforms existing similarity measures, achieving an area under accuracy-efficiency curve (AUC) of 0.8950, compared to 0.8345 for SN-Net and much lower values for MSE, CE, and DM. Notably, CKA, widely used for representational similarity, lags behind KLAS by over 8%. These results along with further experiments in App. D highlight the effectiveness of KL divergence in capturing both representational alignment and functional alignment, making it a superior choice for guiding stitch configuration selection.

| Metric | CKA | MSE | CE | DM | SN-Net | *KLAS (ours)* |
|--------|-----|-----|----|----|--------|--------------|
| **AUC** | 0.8124 | 0.7564 | 0.8023 | 0.7642 | 0.8345 | **0.8950** |

Table 2: Comparison of similarity metrics on the Swin family. Reported values are AUC scores of stitch configuration selection.

### 5.2 ANCHOR SELECTION USING KLAS

We demonstrate that last block KL divergence provides a reliable criterion for selecting pretrained anchors that are more compatible for stitching and more likely to yield better accuracy-efficiency tradeoffs (line 2 in Alg. 3). Since the softmax distribution is already computed in the final blocks of the anchors, additional probes and training are not required at this stage. As shown in Tab. 3 and Fig. 5, for both model families with more than two anchors (i.e., Swin and DeiT), last block KL divergence successfully identifies the anchor pairs that should be stitched together for improved performance.

| Family | Anchors | KL Divergence |
|--------|---------|---------------|
| DeiT | Ti-S
S-B
Ti-B | $4.12 \times 10^{-4}$
**$2.48 \times 10^{-4}$**
$5.62 \times 10^{-4}$ |
| Swin | Ti-S
S-B
Ti-B | $2.70 \times 10^{-3}$
$4.63 \times 10^{-3}$
**$2.38 \times 10^{-4}$** |

Table 3: Last block KL divergence for anchor selection in DeiT and Swin families (pretrained on ImageNet-22K). Lower KL divergence values indicate greater compatibility between anchors for stitching (in blue).

In the Swin family, stitching Ti-B yields a higher AUC compared to Ti-S and S-B stitches chosen by SN-Net. Conversely, within the DeiT family, the Ti-S and S-B stitches selected by SN-Net (via the nearest stitching heuristic) offer a good accuracy-efficiency tradeoff. These trends, as shown in Fig. 5, demonstrate that the nearest stitching heuristic used in SN-Net is not universally optimal across model families. In contrast, selecting anchors based on last block KL divergence offers a principled and generalizable strategy that consistently identifies the most compatible anchor pairs.

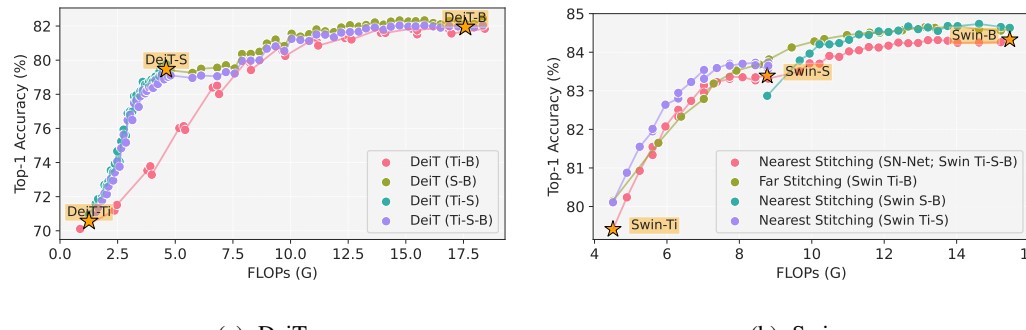

(a): DeiT          (b): Swin

Figure 5: Accuracy-efficiency tradeoffs for anchor selection in DeiT and Swin families. (a) For DeiT, stitching Ti-S and S-B (nearest stitching heuristic from SN-Net) yields the optimal accuracy-efficiency curve, while stitching Ti-B underperforms. (b) Conversely, for Swin, stitching Ti-B (far stitching) provides a superior tradeoff compared to Ti-S-B, which is chosen by SN-Net. These results show that the nearest stitching heuristic is not universally optimal, whereas last block KL divergence (see Tab. 3) consistently identifies the most compatible anchors across families.

## 5.3 BLOCK SELECTION USING KLAS

After selecting the anchors, the next step is to identify block pairs within these anchors for the stitched network finetuning phase. As an initial experiment, we evaluate whether KLAS can automatically recover the paired and unpaired stitching heuristics proposed in SN-Net. To this end, we restrict ourselves to the same anchors used in SN-Net, but perform block selection following lines 4-12 of Alg. 3. Tab. 4 shows that the block selection component of KLAS automatically recovers many of the stitch configurations identified by SN-Net's heuristics, though not perfectly. Crucially, when we finetune stitched networks using the stitches selected by KLAS, our method outperforms SN-Net even under this constrained setting (i.e., using the same anchors as SN-Net), as shown in Fig. 6. A key goal of our KL divergence based approach is to automatically discover stitch configurations that heuristics have shown to be effective in prior work, while also identifying additional promising candidates. The results in Tab. 4 validate KL divergence as a viable similarity metric that can guide stitch discovery without reliance on ad-hoc heuristics.

| Family | Method | Total Stitches | AUC |
|--------|--------|----------------|-----|
| DeiT | SN-Net | 68 | 0.731 |
| | *KLAS* | 66 (40 overlap) | **0.743** |
| Swin | SN-Net | 38 | 0.835 |
| | *KLAS* | 38 (28 overlap) | **0.842** |

Table 4: Comparison of SN-Net and KLAS when constrained to the nearest stitching heuristic (Swin: Ti-S-B; DeiT: Ti-S-B). Despite this restriction, KLAS achieves higher AUC by leveraging its KL divergence based block selection, while also recovering a substantial portion of the heuristic stitches used by SN-Net.

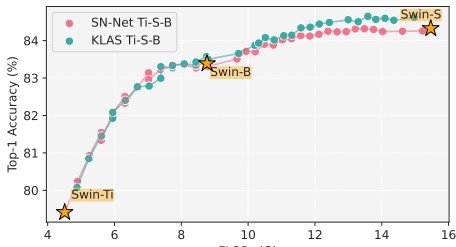

Figure 6: We fix the anchors as the ones selected in the SN-Net (Ti-S-B). We see even then KLAS returns a better accuracy-efficiency curve.

For our main experiments, we follow the setup described in App. B. We use three DeiT anchors Touvron et al. (2021) (DeiT-Ti, DeiT-S, and DeiT-B) and three Swin anchors Liu et al. (2021) (Swin-Ti, Swin-S, and Swin-B). Each DeiT anchor has 12 blocks of equal spatial resolution, resulting in 432 possible stitches across the accuracy-efficiency curve. For Swin, stitching is applied only within stages (as in SN-Net), since spatial resolution is preserved within a stage. Swin-Ti has 12 blocks, while Swin-S and Swin-B have 24 blocks each, leading to 1152 possible stitches in total. We attach probes at the end of every block, yielding 36 probe locations in DeiT and 60 in Swin. KL divergence is then computed for all possible stitch pairs by treating the softmax output at the stitch start point on ImageNet-1K as the source distribution and the corresponding softmax at the stitch end point as the target distribution.

Fig. 7 shows that our algorithm achieves a superior accuracy-efficiency curve across all model families in terms of AUC scores. Even in settings where the nearest stitching heuristic performs

reasonably well (e.g., DeiT), KLAS performs better. We observe that while our algorithm produces a sparser set of stitches at lower FLOPs in Swin, it still closely matches SN-Net in accuracy. The key advantage of our approach is that it does not impose a fixed mapping between source and target blocks; instead, it adapts based on the pretrained anchor weights. Consequently, if the anchor weights change, the selected source-target block pairs may also change. Tab. 4 highlights the overlap between stitches selected by KLAS and those chosen heuristically by SN-Net, showing that KLAS successfully recovers many effective stitches while avoiding rigid reliance on fixed heuristics. A key advancement is that KLAS also discovers new stitch configurations, most notably stitches from Swin-Ti to Swin-B anchors, that were not captured by heuristic approaches in the baseline. We provide further ablation studies on KLAS by comparing it to a naive Min-KL approach in App. C. We discuss KL divergence correlations in App. D, compare KLAS to cascades in App. E and present KL divergence heatmaps and block-level KL divergence variations in App. F.

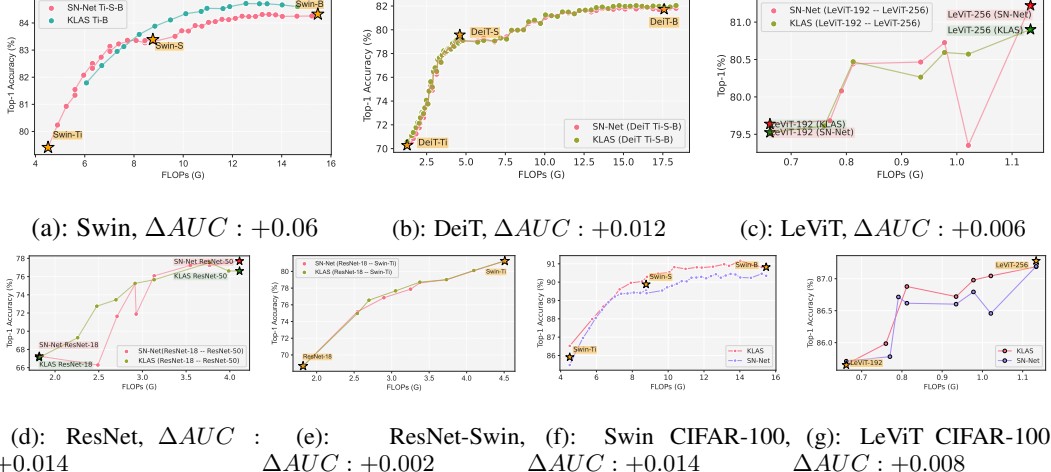

(a): Swin, $\Delta AUC : +0.06$     (b): DeiT, $\Delta AUC : +0.012$     (c): LeViT, $\Delta AUC : +0.006$

(d): ResNet, $\Delta AUC :$ +0.014     (e): ResNet-Swin, $\Delta AUC : +0.002$     (f): Swin CIFAR-100, $\Delta AUC : +0.014$     (g): LeViT CIFAR-100, $\Delta AUC : +0.008$

Figure 7: Accuracy-efficiency tradeoff curves comparing KLAS with SN-Net across multiple model families and datasets: (a) Swin, (b) DeiT, (c) LeViT, (d) ResNet, (e) ResNet-Swin cross-architecture, (f) Swin on CIFAR-100, and (g) LeViT on CIFAR-100. Each point corresponds to a stitched model after finetuning, plotted by FLOPs (x-axis) and Top-1 accuracy (y-axis). Across all settings, KLAS achieves higher accuracy at comparable compute, yielding consistently positive $\Delta AUC$. A positive $\Delta AUC$ indicates that KLAS outperforms SN-Net over the full FLOPs space.

## 5.4 RESULTS ON DENSE PREDICTION TASKS

We further report final semantic segmentation results for Mask2Former Cheng et al. (2022) models with Swin-T and Swin-B backbones on ADE20K Zhou et al. (2019). We assess the stitched models selected by SN-Net and KLAS. All models are evaluated using standard mIoU on the ADE20K validation set; FLOPs are measured per-image with input size $512 \times 512$. The Swin backbones are not frozen in the Mask2Former architecture, while the rest of the model is frozen during finetuning. For each selection strategy, we divide the FLOPs space into three buckets of equal width. The entire stitched network is finetuned for 160k steps under the same schedule and batch size as in the original Mask2Former setup. We use the same steps outlined in §5.1 - §5.3 to run KLAS for the semantic segmentation task. KLAS consistently outperforms SN-Net across the FLOPs space as shown in Tab. 5. For instance, KLAS achieves up to +0.9% mIoU, with comparable FLOPs to the SN-Net

| Model | FLOPs(G) | mIoU(%) |
|---|---|---|
| SN-Net-1 | 152 | 29.4 |
| **KLAS-1** | 145 | **29.8** |
| SN-Net-2 | 274 | 32.6 |
| **KLAS-2** | 277 | **33.5** |
| SN-Net-3 | 327 | 37.7 |
| **KLAS-3** | 316 | **37.8** |

Table 5: Semantic segmentation results on ADE20K using Mask2Former with Swin-T and Swin-B. KLAS-selected stitches outperform SN-Net baselines at similar FLOPs budgets.

counterparts. In general, SN-Net selects lower quality stitch configurations in the same FLOPs buckets. These results confirm that KLAS enables more effective accuracy-efficiency tradeoffs than SN-Net for dense prediction tasks like semantic segmentation. We provide more details on the implementation in App. G.

## 5.5 APPLICABILITY TO LARGE LANGUAGE MODELS (LLMs)

To evaluate whether KLAS generalizes to instruction-tuned LLMs, we replicate a stitching experiment based on ESTA He et al. (2024), comparing it against KLAS-based selection using TruthfulQA Lin et al. (2022). We use Llama 3.2 1B and 3B Grattafiori et al. (2024) as anchor models. Stitched models are created by combining source blocks from Llama 1B to target blocks from Llama 3B. Stitching is done using two strategies: (1) **ESTA**, which selects posi-

| Model | Method | ROUGE-1 | ROUGE-2 |
|---|---|---|---|
| Stitched LLaMa 1.6 B | ESTA | 0.576 | 0.304 |
| Stitched LLaMa 1.4 B | **KLAS** | **0.593** | **0.337** |
| Stitched LLaMa 2.7 B | ESTA | 0.631 | 0.353 |
| Stitched LLaMa 2.6 B | **KLAS** | **0.645** | **0.379** |

Table 6: Results on LLMs using Llama 1B and 3B. KLAS-selected stitched models improve over ESTA-selected stitched models, with fewer parameters.

tions based on layer count (e.g., 1B first $k$ layers + 3B remaining), and (2) **KLAS**, which selects based on minimum stitch score using per-layer next-token distributions using linear probes. Evaluation is performed in the generative setting on TruthfulQA, and we report *ROUGE-1* and *ROUGE-2* scores against reference completions after finetuning for 10 epochs in both strategies. Tab. 6 shows that KLAS consistently identifies more similarly aligned stitch points (i.e., blocks) than ESTA's heuristic-based selection. On TruthfulQA, KLAS-stitched models outperform similar-sized ESTA-stitched models by up to 0.017 ROUGE-1 and 0.033 ROUGE-2, thus offering a more principled and effective approach to designing stitched LLMs. We provide more experimental details in App. H.

## 5.6 ABLATION STUDIES ON KLAS HYPERPARAMETERS

**Threshold ($\tau$).** KLAS selects stitch configurations by thresholding the stitch score $\Gamma(i, j)$ relative to the minimum score within each FLOPs bucket. We sweep $\tau$ from 1% to 10% more than this per-bucket minimum and report the resulting average accuracy and AUC in Tab. 7. A clear tradeoff emerges: very high thresholds (e.g., $\tau = 10\%$) admit noisy, low-quality stitch configurations, while low thresholds (e.g., $\tau = 1\%$) become overly selective and reduce the number of selected stitches, yielding sparse Pareto fronts. The default $\tau = 5\%$ provides the best balance between robustness and diversity. We use Swin-Ti, Swin-S and Swin-B with 20 buckets for these experiments.

| Threshold ($\tau$) | Avg Top-1(%) | AUC |
|---|---|---|
| 1% | 83.72 | 0.8934 |
| 3% | 83.74 | 0.8942 |
| 5% | **83.76** | **0.8950** |
| 10% | 83.69 | 0.8931 |

Table 7: Effect of threshold $\tau$ on KLAS stitch selection. High $\tau$ admits noisy pairs; low $\tau$ causes sparsity.

**Bucket Granularity.** To understand the effect of discretizing the FLOPs space, we vary the number of FLOPs buckets used during stitch selection. The block granularity is inversely proportional to the number of buckets. As shown in Tab. 8, using finer buckets (i.e., lower bucket granularity) increases the density of Pareto front, but KLAS maintains strong performance even with coarser buckets (i.e., higher bucket granularity). Overall sensitivity remains low for any reasonable bucket count (10-20). We use Swin-Ti, Swin-S and Swin-B with $\tau = 5\%$ for these experiments.

| #Buckets | Avg Top-1(%) | AUC |
|---|---|---|
| 10 | 83.65 | 0.8902 |
| 15 | 83.75 | 0.8947 |
| 20 | **83.76** | **0.8950** |

Table 8: Ablation on FLOPs bucket granularity. KLAS remains robust across bucket granularity choices.

## 6 DISCUSSION AND CONCLUSION

This paper introduces KLAS, a novel, efficient and automatic framework for improving the accuracy-efficiency tradeoff produced by stitching pretrained models. The method leverages KL divergence to identify stitched configurations that are likely to perform well after finetuning the stitched super-network. Empirical evaluations show that KLAS consistently outperforms heuristic-based baselines by providing a principled, similarity-driven selection strategy that generalizes across diverse model architectures, datasets, tasks and modalities, while incurring negligible additional computational cost. Beyond accuracy alone, generative workloads impose distinct system constraints across the prefill and decode phases (e.g., *time-to-first-token* and *time-between-tokens*). This creates a new optimization problem: balancing accuracy with two latency budgets (and often memory / throughput constraints) across both phases. A natural future direction is to extend KLAS to this setting by enforcing KV-cache compatibility across stitched blocks and redefining similarity measures accordingly.

ACKNOWLEDGMENTS

This material is based upon work partially supported by the National Science Foundation (NSF) under Grant Number CNS-2420977 as well as sponsored research awards from Cisco Research and Georgia Tech Research Institute (GTRI). This work was further supported in part by CoCoSys, one of seven centers in JUMP 2.0, a Semiconductor Research Corporation (SRC) program sponsored by DARPA. We would also like to express our sincere gratitude to the reviewers and the AC panel for their insightful comments and thoughtful consideration. Disclaimer: Any opinions, findings, and conclusions or recommendations expressed in this material are those of the authors and do not necessarily reflect the views of the National Science Foundation.

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

## A  ALGORITHMS

---

**Algorithm 1** ProbeNet Pseudocode (see §4.1)

---

1: **Class** ProbeNet
2:   **Init**(model, stage_block_info, output_dim = 1000):
3:     Store $model$ as anchor (frozen)
4:     **for each** stage in stage_block_info:
5:       **for each** block_dim in stage:
6:         Create probe: Linear($block\_dim \rightarrow output\_dim$) + Softmax
7:         Store probe in self.layers[stage][block]
8:     Set current_features ← None
9:     Freeze anchor parameters; keep probes trainable
10:
11:   **Method** extract_block_features(x):
12:     Run anchor.extract_block_features(x) with no_grad
13:     Store result in current_features
14:
15:   **Method** clear_block_features():
16:     Set current_features ← None
17:
18:   **Method** iter_blocks():
19:     **yield** (stage_id, block_id) for each stage/block
20:
21:   **Method** forward(x, stage_id, block_id, get_all_probe_losses, criterion, target):
22:     **if** get_all_probe_losses:
23:       return anchor.get_probe_losses(x, self.layers, criterion, target, dims)
24:     **if** current_features is None:
25:       extract_block_features(x)
26:     Get block feature $\rightarrow$ reshape $\rightarrow$ pass through probe
27:     **return** output

---

**Algorithm 2** Get Similarity Scores (see §4.2)

---

**Inputs:** Set of all anchor models $M$
**Output:** Set of all similarity scores $\mathcal{K}$

---

1: $\mathcal{K} \leftarrow \{\}$
2: **for** each model $m$ in $M$ **do**
3:     **for** each block $i$ in $m$ **do**
4:         Add a linear classifier probe $P_i^m$
5:         Train $P_i^m$ on train split $(\mathcal{D}_t)$
6:     **end for**
7: **end for**
8: **for** each model pair $(m, n)$ in $M$, with $m$ having lower complexity than $n$ **do**
9:     **for** each block $i$ in anchor $m$ **do**
10:         **for** each block $j$ in anchor $n$ **do**
11:             $\mathcal{K} \leftarrow \mathcal{K} \cup \{\Theta(P_i^m, P_j^n)\}$
12:         **end for**
13:     **end for**
14: **end for**
15: Return $\mathcal{K}$

---

## B  EXPERIMENT SETUP

**Baselines.**  We first compare KLAS with existing stitching methods for state-of-the-art image classification models w.r.t. accuracy-efficiency tradeoffs. Specifically, we compare KLAS with the

---

**Algorithm 3** KLAS: KL divergence based Anchor Stitching (see §4.2)

---

**Inputs:** Anchor models $M$, similarity scores $\mathcal{K}$ from Alg. 2, threshold $\tau$, buckets $\mathcal{B}$
**Output:** Set of selected stitch configurations $\mathcal{S}$

1: $\mathcal{S} \leftarrow \{\}$
2: Pick anchors $f, g \in M$ based on their final layer KL divergence
3: Sort all the stitches based on FLOPs
4: **for** each bucket $b \in \mathcal{B}$ **do**
5:     $\mathcal{R}_b \leftarrow \{\}$
6:     **for** each stitch configuration $(i, j) \in b$ **do**
7:         Compute $\Gamma(i, j)$ (see Eq. 2)
8:         $\mathcal{R}_b \leftarrow \mathcal{R}_b \cup \Gamma(i, j)$
9:     **end for**
10:    Compute set of stitch configs $\mathcal{R}_b^*$ using $\mathcal{R}_b$ (see Eq. 3)
11:    $\mathcal{S} \leftarrow \mathcal{S} \cup \mathcal{R}_b^*$
12: **end for**
13: Return $\mathcal{S}$

---

primary baseline SN-Net Pan et al. (2023), which introduced heuristic stitching techniques including nearest stitching and paired / unpaired stitching. The finetuning costs of all techniques are *50 epochs* and we use all the same parameters and hyperparameters for finetuning as in Pan et al. (2023) to enable a fair comparison. We run all experiments on $8{\times}$A40 NVIDIA GPUs.

**Success Metric.** KLAS is compared against the SN-Net baseline on the following success metric, *accuracy-efficiency tradeoff*: accuracy of selected stitched models as a function of FLOPs. To compare tradeoff spaces obtained from different baselines, we use the *area under the curve* (AUC) of accuracy and normalized FLOPs. The higher the AUC, the better. Additionally, this metric measures the stitch configuration quality by evaluating how effectively each method selects stitch points that are likely to perform well.

**Datasets.** We evaluate all methods on CIFAR-100 and ImageNet-1k datasets. The complexity of datasets increases from CIFAR-100 to ImageNet-1k. The datasets vary in the number of classes, image resolution, and number of train / test samples. The stitching models and the probes were trained on the train splits and the KL divergence values in Alg. 2 were obtained using the validation splits of the datasets. The train and validation splits were obtained from the official train set of ImageNet-1K. We report the final accuracies on the official validation set of ImageNet-1K. We use FLOPs as a proxy for the efficiency of the deployment point (i.e., stitched model), as is widely used in NAS works Oygenblik et al. (2025); Annavajjala et al. (2024); Khare et al. (2024); Sanyal et al. (2023).

**Model Architecture Families.** All experiments are conducted using pretrained models from several different architecture families: transformers including DeiT Touvron et al. (2021), Swin Liu et al. (2021), LeViT Graham et al. (2021), CNNs including ResNet He et al. (2016), and LLMs including Llama Grattafiori et al. (2024). All anchor models are pretrained on a variant of the dataset being used for finetuning the stitched network using standard training protocols outlined in Pan et al. (2023). We use the ResNet and Swin architectures for our experiments on cross-architecture stitching. To ensure fair comparison, KLAS and all baselines use identical pretrained weights for the corresponding anchor models, and all stitching configurations are evaluated using the same batch sizes as in Pan et al. (2023).

**Hyperparameters.** The hyperparameters for each model were set as defined in their respective papers (Liu et al., 2021; Touvron et al., 2021; Graham et al., 2021; He et al., 2016). The stitched network finetuning hyperparameters were adapted from the SN-Net codebase ZipLab (2023). We set the threshold ($\tau$) to 5% more than the minimum stitch score ($\Gamma$) in the bucket. We assign the bucket granularity used for creating the set of buckets ($\mathcal{B}$) such that every block in the target anchor $g$ has at least one representative stitch configuration in $\mathcal{S}$. This gives 20 buckets for the Swin model family.

## C    MIN-KL APPROACH VS. KLAS

While KL divergence is a useful metric for identifying promising stitches, using a Min-KL strategy of directly selecting the stitches with the lowest KL values and training them is suboptimal. The key issue is that KL divergence is not symmetric, and stitches with different target blocks are not directly comparable. To address this, stitches must be evaluated in a target-aware manner. KLAS addresses this by selecting the stitch with the lowest similarity score $\Gamma$ (see Eq. 2) for each bucket, ensuring both fairness across buckets / target blocks and balanced coverage of stitched models across the FLOPs range. As shown in Fig. 8, this principled strategy outperforms the naive Min-KL baseline, which underperforms even compared to SN-Net on DeiT and Swin.

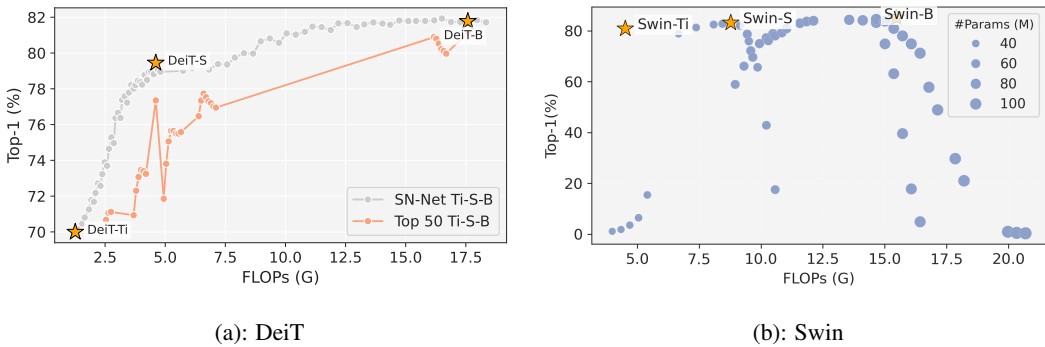

(a): DeiT                                    (b): Swin

Figure 8: Min-KL algorithm is suboptimal because naively choosing the stitch configurations with the lowest KL divergence values is not target-aware across the FLOPs space. KL divergence is not a symmetric metric and hence all candidate stitch configurations across the FLOPs space cannot be compared to each other directly.

## D    A DEEPER DIVE INTO KL DIVERGENCE

### D.1    WHY KL OUTPERFORMS MSE, CE, CKA, AND DM FOR STITCH SELECTION

We attribute KL divergence's superior stitch configuration recovery rate (Tab. 1 shows it recovers 88.9% of viable Swin Ti-B stitch configs vs. 22-33% for MSE/CE/DM and only 5.5% for CKA) to its ability to capture both *representational* and *functional* alignment. Let $f_{\leq i}(x)$ and $g_{\leq j}(x)$ be intermediate features from source and target models. Let $P_i^f(x)$ and $P_j^g(x)$ be softmax probability distributions from linear probes. The similarity metrics are defined as follows:

$$\text{MSE:}\quad \frac{1}{|D_v|}\sum_x \|f_{\leq i}(x) - g_{\leq j}(x)\|^2$$

$$\text{CE:}\quad -\sum_x \log P_i^f(x)[y(x)]$$

$$\text{DM:}\quad \min_T \sum_x \|T \circ f_{\leq i}(x) - g_{\leq j}(x)\|^2$$

$$\text{CKA:}\quad \frac{\|F^\top G\|_F^2}{\|F^\top F\|_F \cdot \|G^\top G\|_F}$$

| Metric | Pearson $r$ (to $\Delta$Acc) | Spearman $\rho$ |
|---|---|---|
| Stitch score ($\Gamma$) | **0.84** | **0.81** |
| MSE | 0.42 | 0.37 |
| CKA | 0.28 | 0.21 |

Table 9: Pearson and Spearman correlations between candidate similarity metrics and stitched model accuracy drop ($\Delta$Acc). The stitch score $\Gamma$ (based on KL) exhibits the strongest predictive correlation, confirming its utility for selecting performant configurations (evaluated on Swin family).

To measure how well different similarity metrics predict stitch configuration quality, we compute how each metric score relates to the final stitched model's accuracy. For every stitch pair, we calculate the accuracy drop compared to the target model as $\Delta\text{Acc} = \text{Accuracy}_{\text{target}} - \text{Accuracy}_{\text{stitched}}$. We then compute two correlation scores between the similarity metric and $\Delta\text{Acc}$: **Pearson correlation** ($r$) measures linear relationship, and **Spearman correlation** ($\rho$) measures whether ranks agree (i.e., monotonic trend). Results are shown in Tab. 9. The stitch score ($\Gamma$ from Eq. 2) (computed using KL divergence between probe distributions) has the strongest correlation with accuracy drop, indicating that stitch candidates with low stitch score are likely to perform better after finetuning. In contrast, MSE and CKA show much weaker correlation. This helps us conclude that $\Gamma$ shows a near-monotonic increase: as the score rises, so does the accuracy drop, while CKA and MSE exhibit noisier and less predictive patterns.

### D.2 KL DIVERGENCE IS BOTH REPRESENTATIONAL AND FUNCTIONAL

To show that the KL divergence based stitch score ($\Gamma$) captures both representational and functional similarity, we run two additional experiments:

**1. Shuffle-label Stitch Score:** We retrain the probes with randomly shuffled labels. This removes class semantics while preserving feature structure. The resulting scores lose correlation with task performance, similar to unsupervised metrics.

**2. Class-Conditional CKA:** We compute CKA separately per class, then average across classes to inject label information. This improves correlation modestly but still underperforms w.r.t. stitch score. The stitch score outperforms other metrics because it reflects both representational structure and functional behavior, as shown in Tab. 10. Supervision is essential; without it, metric utility drops to the level of unsupervised baselines.

| Variant | Pearson $r$ | Spearman $\rho$ |
|---|---|---|
| Stitch score ($\Gamma$) | **0.84** | **0.81** |
| Shuffled-label score | 0.19 | 0.16 |
| CKA (vanilla) | 0.28 | 0.21 |
| CKA (class-conditional) | 0.51 | 0.46 |

Table 10: Correlation of stitch performance with label-aware and label-free metrics. Removing supervision (shuffled labels) weakens KL-based stitch score, while adding class-conditioning to CKA improves its stitch ranking. These results confirm that supervision is critical for good stitch selection.

## E    KLAS AND SN-NET VS. MODEL CASCADES

We compare KLAS and SN-Net to early-exit cascaded inference baselines adapted from Lebovitz et al. (2023). Since in our work we only consider binary stitches, in our setup, we have just two starting models. Inference begins with a small model (Swin-S), and proceeds to a larger model (Swin-B) only if the small model's confidence (maximum softmax) falls below a threshold. Thresholds are tuned to optimize the accuracy-FLOPs Pareto front, and the average inference cost is measured in FLOPs over the full ImageNet-1K validation set. The cascaded models are not finetuned and share no parameters. For our experiment, we use their method with the following settings: (i) input resolution is $224 \times 224$ and (ii) routing is based on the max softmax score. We sweep thresholds in the range $[0.6, 0.95]$ and report the best value per FLOP bin. For example, thresholds of 0.85, 0.90, and 0.95 correspond to mean budgets of 10.1G, 11.3G, and 12.6G FLOPs, respectively.

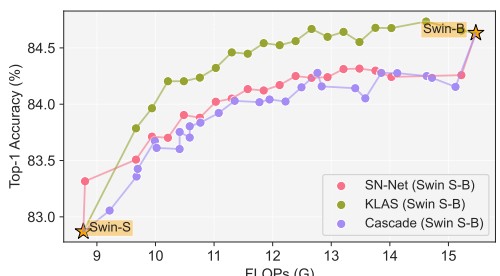

Figure 9: Accuracy-FLOPs tradeoff on ImageNet-1K for Swin-S and Swin-B. KLAS and SN-Net stitched models outperform cascaded inference baselines that use softmax-threshold early exits. FLOPs for cascades are *average* inference cost over the validation set. FLOPs for KLAS are deterministic by design.

As shown in Fig. 9, both KLAS and SN-Net produce stitched models that are consistently superior to cascaded inference across the full FLOPs spectrum. For instance, at >12 GFLOPs, KLAS improves over the best cascade by >0.4% Top-1 accuracy. This trend holds from low to high FLOPs regimes. Despite the adaptive nature of cascades, KLAS-based stitched models yield stronger Pareto front performance with two starting models for two reasons: (1) stitching enables model restructuring by stitching blocks with similar learned representations into a unified network, rather than deferring work across the smaller and larger models; and (2) the well-known problem with cascades that samples are routed based on softmax confidence, which is poorly calibrated in low-entropy regions, leading to suboptimal exits Wang et al. (2020); Lebovitz et al. (2023). In contrast, stitching builds a single, end-to-end architecture that avoids runtime uncertainty, yielding higher accuracy per FLOP even in the average-case, as shown in Fig. 9. It is easy to see that in the worst-case scenario, stitching will perform better than cascades for this particular setting (i.e., with two models).

We further evaluate a *cross-family* pair consisting of a CNN and a Transformer: ResNet-18 (1.8 GFLOPs) and Swin-B (15.4 GFLOPs). We construct three KLAS stitched configurations whose FLOPs are close to those of three corresponding early-exit cascades, tuned with the same softmax-threshold procedure as in Lebovitz et al. (2023). Tab. 11 shows that even when the two anchors are architecturally *different*, KLAS consistently outperforms cascaded inference at similar or lower FLOPs, improving Top-1 accuracy by 2-3% across all three operating points.

| Config | Method | FLOPs (G) | Top-1 (%) |
|---|---|---|---|
| 1 | KLAS | **3.409** | **79.60** |
| | Cascade | 3.455 | 76.13 |
| 2 | KLAS | **15.043** | **84.52** |
| | Cascade | 15.197 | 82.46 |
| 3 | KLAS | **17.127** | **84.74** |
| | Cascade | 17.183 | 82.79 |

Table 11: KLAS vs. early-exit cascades for a cross-family pair (ResNet-18 → Swin-B) on ImageNet-1K. KLAS outperforms cascaded inference Lebovitz et al. (2023) at similar or slightly lower FLOPs across all three operating points. FLOPs for cascades are *average* inference cost over the validation set. FLOPs for KLAS are deterministic by design.

We would like to further draw attention to the fact that the FLOP numbers reported for cascades in Fig. 9 and Tab. 11 correspond to *average cost* over the validation set. The compute of a cascade is not deterministic: it depends on how often examples fall below the confidence threshold and are forwarded to the second model. In practice, many inputs remain below the threshold, so cascades execute all the blocks of the first anchor and all the blocks of the second one. This often yields only modest accuracy gains while spending extra FLOPs on unnecessary blocks. By contrast, once a KLAS stitched configuration is selected and finetuned, the model executes a fixed sequence of blocks for every input, leading to a deterministic and easily budgetable FLOP cost for all samples.

The KL divergence heatmaps in Fig. 10 further clarify why stitching selects different configurations than cascades. A pure cascade that always runs the full small model and then the large model corresponds to connecting the *last* block of the small anchor to the *first* block of the large anchor, i.e., the upper-left corner in each heatmap. These block pairs have among the highest KL divergence values, indicating poor representational and functional compatibility. KLAS explicitly searches for low KL divergence pairs and therefore prefers connections in the darker regions near the diagonal. As a result, such "top-left" cascade-style configurations are never chosen: they are dominated by lower KL divergence stitches that achieve better accuracy-FLOPs tradeoffs than cascades.

## F  KL DIVERGENCE HEATMAPS

KL Divergence heat maps provide a visual representation of the similarity between intermediate blocks of two anchor models. Each entry in the heat map corresponds to the KL divergence between the softmax distributions of a source block and a target block, as measured by linear probes. Low values (darker regions) indicate block pairs with higher compatibility for stitching, while higher

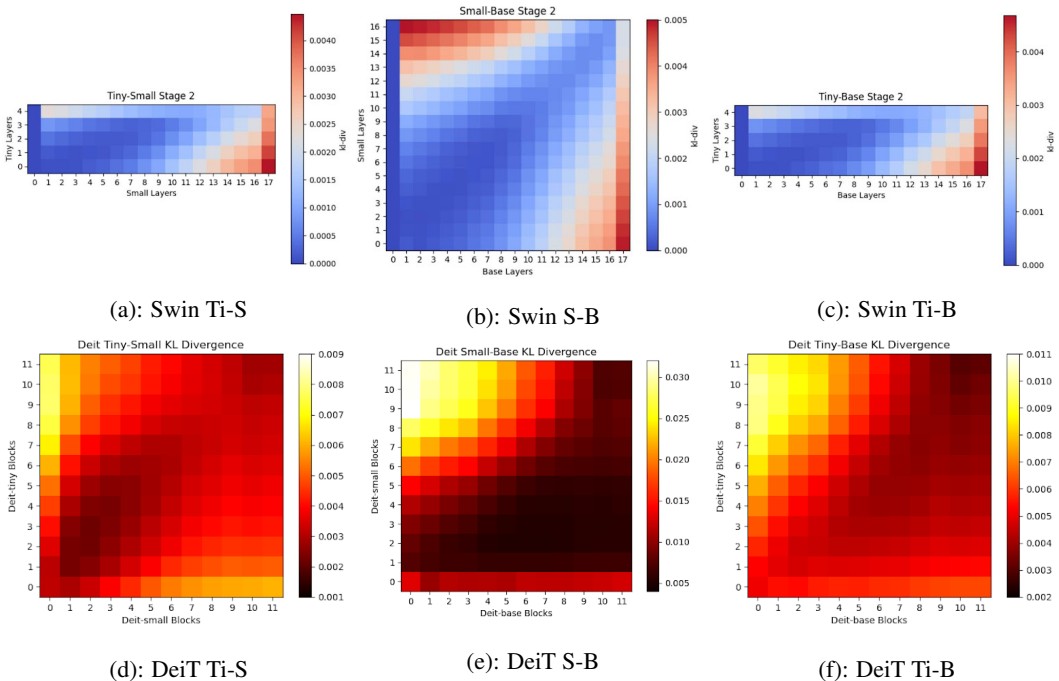

Figure 10: Heatmaps of KL divergence values across different block pairs for Swin Stage 2 (top row) and DeiT (bottom row) anchor combinations. Each cell represents the divergence between the output probability distributions of a source block (x-axis) and a target block (y-axis). Lower values (cooler colors) indicate higher compatibility between block pairs, making them more promising candidates for stitching.

values suggest poor alignment. These heat maps allow us to identify promising stitch configurations at a glance.

In Fig. 10, we observe that the KL divergence values are generally lower when stitching between blocks of similar depth across anchors, as shown by the darker regions near the diagonals in most heatmaps. This indicates that earlier blocks align better with earlier blocks and later blocks with later blocks. However, this does not *exactly* give us paired stitching, a heuristic used in SN-Net. Additionally, in both Swin and DeiT, larger anchors (e.g., Base) tend to introduce higher divergence when paired with earlier blocks of smaller anchors, highlighting their greater representational shift.

Fig. 11 visualizes block-level KL divergence across different anchor pairs in Swin Ti-B Stage 2. A clear trend emerges: divergence is lowest along regions where blocks of similar depth are compared, suggesting that early-to-early and late-to-late block pairings are naturally more compatible for stitching. In contrast, pairing shallow blocks from smaller anchors with deeper blocks from larger anchors results in higher divergence, reflecting greater representational mismatch. These structured patterns validate that KL divergence effectively captures cross-anchor compatibility and provides a principled signal for selecting good block pairs.

## G   DETAILS ON THE SEMANTIC SEGMENTATION EXPERIMENT

A key difficulty arises from the architectural expectations of dense prediction models, which typically rely on backbones such as ViTs and ResNets. These models are integrated with additional components (e.g., pixel decoders, transformer decoders, detection heads) that require activations of specific dimensions Cheng et al. (2022); Ravi et al. (2024); Redmon et al. (2016). In contrast, SN-Net introduces stitched backbones and anchors with varying intermediate activation sizes, which creates a mismatch in feature dimensions and prevents straightforward training. We address this activation-dimension inconsistency using stage-specific linear maps. Promising directions include the design of flexible mapping layers or adaptive projection mechanisms that can align heterogeneous feature representations across stitch configurations. Another avenue is to explore more efficient training

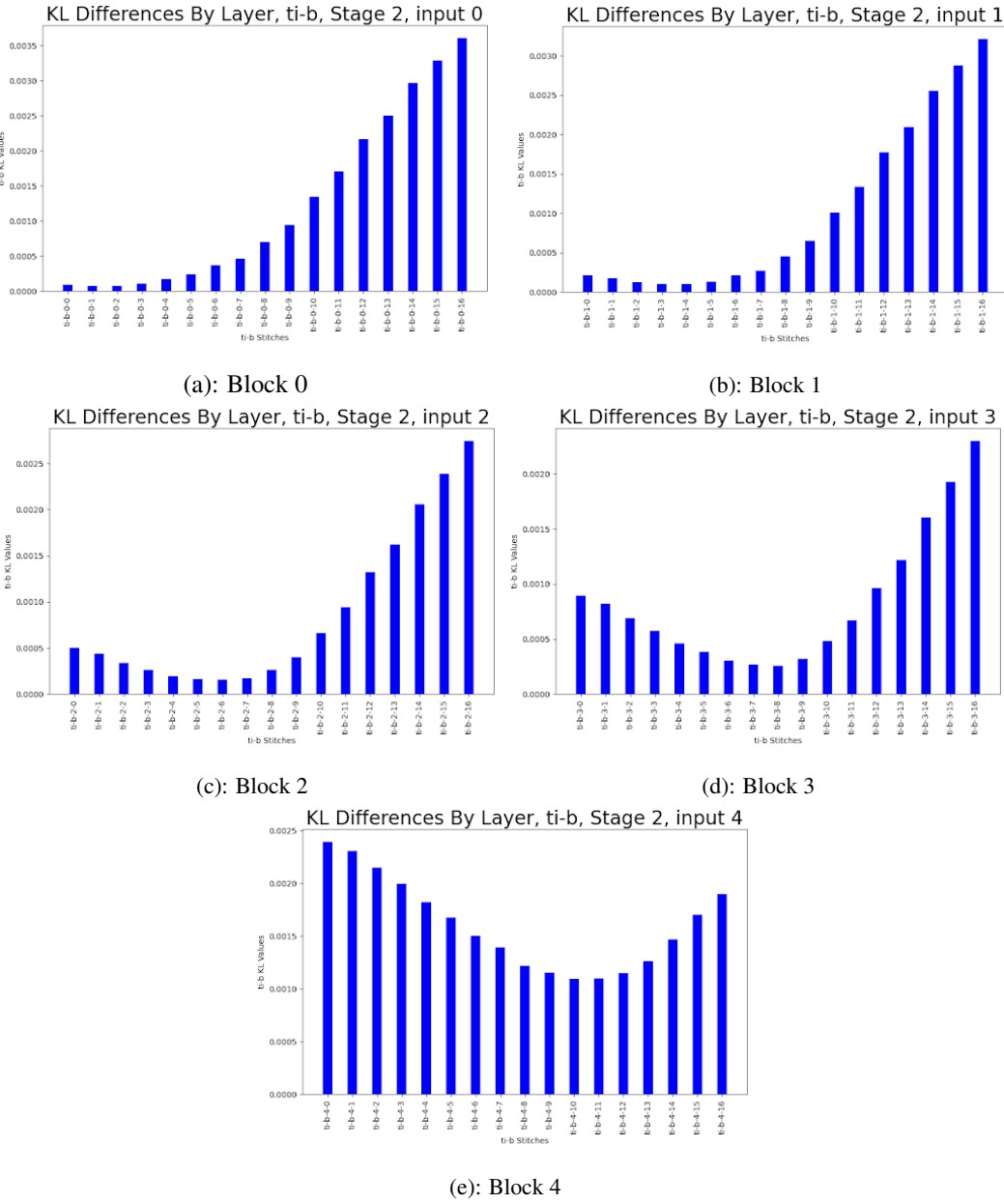

Figure 11: KL divergence values across different target layers when stitching from Swin-Ti to Swin-B in Stage 2. Each sub-figure shows the KL divergence distribution for one source block (Blocks 0-4) with different target block inputs. The plots highlight that KL divergence varies systematically across layers, with certain source-target block pairs yielding significantly lower divergence values, indicating better stitching compatibility.

strategies that minimize additional computational overhead while maintaining compatibility with downstream dense prediction heads.

The original M2F architecture uses a hierarchical transformer-based backbone, where feature maps from each stage are fed into a pixel decoder and a transformer decoder. When replacing the backbone with a stitched Swin backbone, the M2F architecture expects feature maps with consistent dimensions. However, since stitched networks combine layers from multiple Swin variants (e.g., Tiny, Small, Base), their intermediate feature dimensions may vary across stages in the models. This necessitates careful design modifications to ensure compatibility. Our stitching and finetuning procedure is as follows:

1. Stitch layers initialized using pseudo-inverse based on source & target feature dimensions.

2. Four additional linear transformation layers (randomly initialized) are inserted (one per stage) to project stitched outputs to match the expected shape of Swin-S or Swin-B at each stage, since only these two can be target anchors as we fix the direction in stitching.

3. The M2F head (pixel + transformer decoder) is kept frozen during training to minimize overhead.

4. On each mini-batch, a random stitch configuration is sampled.

5. Full forward pass through the SN-Net backbone and M2F head.

6. Compute segmentation loss (cross entropy + auxiliary losses).

7. Update backbone + linear mapping weights based on loss gradients.

There is a key challenge in enabling dimensionally consistent feature flow from stitched backbone to decoders. This is traced to the fact that stitch configurations do not always end with the same final feature dimensionality (e.g., 384, 768, or 1024 channels depending on the final layer in the sampled path). The decoders expect fixed final feature dimensions. We solve this using stage-specific linear maps, which are randomly initialized.

## H  DETAILS ON THE LLM STITCHING EXPERIMENT

For all LLM stitching experiments in §5.5, we adopt an identical finetuning setting derived from the ESTA setup He et al. (2024) to ensure that the two methods are comparable. We stitch between Llama 3.2 instruction-tuned 1B and 3B anchors Grattafiori et al. (2024) and evaluate performance in the generative setting on TruthfulQA benchmark Lin et al. (2022). All stitched models in both ESTA and KLAS are trained under identical hyperparameters: 10 epochs finetuning of next-token prediction with cross-entropy loss, batch size 64, learning rate $5 \times 10^{-5}$ with cosine decay, a 1-epoch warmup period, AdamW optimizer, and a 128-token sequence length.

The training and test datasets are *not* identical. We use the generative setting of TruthfulQA benchmark and apply a 60/40 split: (i) finetuning (training) is performed on 60% of TruthfulQA, and (ii) evaluation (test) is performed exclusively on the held-out 40%. All stitched models (both ESTA and KLAS) are finetuned only on the 60% portion and are never exposed to the remaining 40% used for evaluation. Thus, all stitched LLMs are evaluated strictly on unseen TruthfulQA questions. We also note that the Llama 3.2 instruction-tuned 1B and 3B anchor models Grattafiori et al. (2024) are pretrained on a large mixture of web-scale English corpora, code, and instruction-following datasets, and do not include TruthfulQA in their pretraining mix.

ESTA applies fixed stitching points (e.g., using the first $k$ layers of the 1B model and the remaining layers from the 3B model), whereas KLAS selects stitch locations using KL divergence computed over linear probe outputs to select stitch points that have similar next-token prediction distributions. Tab. 6 shows that across comparable stitched model sizes, KLAS consistently yields higher ROUGE-1 and ROUGE-2 scores than ESTA, demonstrating better stitch configuration selection and hence, an improved Pareto front of stitched models.

## I  USE OF LLMS IN OUR SUBMISSION

Yes, we used Large Language Models (LLMs) to aid in polishing the writing. Specifically, LLMs were used for grammar checking, improving readability, and refining phrasing. All technical content, experiments, and analyses were conducted and written by the authors.

