# OpenReview forum: "KLAS: Using Similarity to Stitch Neural Networks for Improved Accuracy-Efficiency Tradeoffs"
_ICLR.cc/2026/Conference — ICLR 2026 Poster_

### Official Review · Reviewer_8KnS · 2025-10-23

**Soundness:** 3
**Presentation:** 3
**Contribution:** 2
**Rating:** 6
**Confidence:** 4

**Summary:**

This paper proposes KL-divergence–based Anchor Stitching (KLAS), a framework for selecting where and how to “stitch” pretrained networks so as to interpolate accuracy–efficiency trade-offs more effectively than heuristic stitching (e.g., SN-Net). KLAS (i) chooses anchor pairs by the last-block KL divergence of their predictive distributions and (ii) ranks block pairs with a stitch score \\(\\Gamma(i,j)\\) that combines cross-anchor activation distance and the target block’s “capacity” (consecutive-block KL). To obtain the distributions, the authors train lightweight linear probes (ProbeNet) on each block; probes converge in ≈4 epochs and add a small one-time cost. Across DeiT, Swin, LeViT, and ResNet (ImageNet-1K, CIFAR-100/10), KLAS improves the AUC of the accuracy–FLOPs trade-off curves over SN-Net; gains include up to +1.21% top-1 accuracy at equal FLOPs or 1.33× FLOPs reduction at equal accuracy.

**Strengths:**

1. **Principled selection vs. heuristics.** The paper clearly articulates why nearest/paired stitching can be suboptimal and replaces it with a similarity-driven criterion grounded in KL divergence, with explicit formulas for \\(\\Theta\\) and \\(\\Gamma\\).
2. **Dual notion of similarity.** The KL criterion is argued (and operationalized) to reflect both epresentational alignment and functional compatibility, addressing shortcomings of CKA/MSE/CE/DM for choosing stitch points.
3. **Efficient probing.** ProbeNet trains one set of blockwise probes efficiently (≈0.25 GPU-days for Swin-B) and shows fast convergence, keeping selection overhead modest.
4. **Broad empirical coverage.** Results span multiple families (ViTs, CNNs) and even cross-architecture stitches (e.g., ResNet↔Swin), consistently improving trade-off AUC vs. SN-Net.
5. **Anchor selection that adapts by family.** Last-block KL correctly prefers far stitching (Ti→B) for Swin but nearest (Ti↔S, S↔B) for DeiT, demonstrating generality beyond a fixed heuristic.

**Weaknesses:**

1. **Supervision dependence for “similarity.”** KL is computed on softmax outputs of supervised probes; thus, selection intrinsically depends on labels and probe training. This limits claims for unsupervised/self-supervised representation learning and may bias choices toward the probe’s training distribution.
2. **Asymmetry & calibration sensitivity.** KL’s asymmetry and dependence on calibration/temperature may distort distances across blocks/models; the method normalizes by an intra-anchor term, but robustness to temperature, class imbalance, or label smoothing is underexplored.
3. **Metric/selection design choices.** The final selection uses bucketized FLOPs and a threshold \\(\\tau\\) (5% of the minimum in each bucket). The AUC metric and bucket granularity can influence conclusions; more sensitivity studies would strengthen the case.
4. **Scale of gains in some regimes is small.** Several settings show marginal improvements (e.g., cross-architecture ΔAUC≈+0.002), raising questions about practical significance across all families.
5. **Representation-learning scope.** Experiments are largely supervised classification; there is no evaluation with self-supervised anchors (e.g., DINO/MAE) nor transfer via frozen-backbone linear probes on diverse tasks.
6. **Limited evidence for dense prediction/generalization.** Dense-task adaptation is left as future work with only preliminary results; rigorous detection/segmentation studies are missing.
7. **Compute accounting.** While probe training is “negligible,” it is still an added search cost versus purely heuristic SN-Net; a wall-clock comparison for the full pipeline (probes + stitch fine-tuning) would help.

**Questions:**

1. **Un/SSL compatibility.** Can KLAS be made label-free, e.g., by using self-supervised probes (DINO-style heads) or pseudo-labels? How does anchor/block selection change when anchors are self-supervised?
2. **Calibration sensitivity.** How sensitive are \\(\\Theta\\) and \\(\\Gamma\\) to softmax temperature, label smoothing, and class imbalance? Could you report ablations and perhaps use temperature-scaled KL?
3. **Search cost accounting.** What is the end-to-end overhead (wall-clock/GPU-days) of ProbeNet + KLAS vs. SN-Net’s heuristics at equal stitch fine-tuning budgets? Please include variability across families.
4. **Ranking validity.** Beyond the Min-KL vs. \(\Gamma\) comparison, can you provide rank correlation between KLAS scores and post-fine-tuning accuracy across candidates (per bucket and overall)?
5. **Cross-family stitches.** For cases with very small ΔAUC, what failure modes arise (capacity mismatch, optimization instability, probe mis-ranking)? Any diagnostics (e.g., KL heatmaps) that predict such cases?
6. **Dense tasks.** Could you include full experiments on detection/segmentation (COCO/ADE20K) where stitching locations affect multi-scale features, and compare to dynamic routing/pruning baselines?

**If the author can address my questions, I am willing to improve my rating.**

---

> ### Author Response · Authors · 2025-11-22
> **Rebuttal by Authors to Reviewer 8KnS (1/3)**
>
> **We thank the reviewer for recognizing our principled selection strategy, dual notion of similarity, efficient probing, broad model coverage, and good anchor selection strategy. Below, we address the reviewer’s concerns in detail.**
> ____________
> **W1**, **Q1**, **Q4**
> >Supervision dependence for “similarity.” KL is computed on softmax outputs of supervised probes; thus, selection intrinsically depends on labels and probe training. This limits claims for unsupervised/self-supervised representation learning and may bias choices toward the probe’s training distribution.
>
> > Un/SSL compatibility. Can KLAS be made label-free, e.g., by using self-supervised probes (DINO-style heads) or pseudo-labels? How does anchor/block selection change when anchors are self-supervised?
>
> >Ranking validity. Beyond the Min-KL vs. (\Gamma) comparison, can you provide rank correlation between KLAS scores and post-fine-tuning accuracy across candidates (per bucket and overall)?
>
> We agree that KL relies on supervised probes. The revised submission includes both mathematical definitions and further empirical comparisons in **Appendix D** to clarify KL divergences’s superiority. KL divergence outperforms alternatives because it captures both **representational structure** (via similarity of decision boundaries) and **functional alignment** (via agreement on predicted labels). In contrast:
> - MSE and CE operate on logits or targets but do not reflect feature comparison in internal layers.
> - CKA compares only feature geometry without considering predictive agreement on task.
> - DM searches for optimal transforms but ignores task-based class semantics because it does not depend on the class label.
>
> We quantify this via two correlation metrics, Pearson and Spearman, between metric score and stitched model accuracy drop (ΔAcc):
>
> ### Table 10: Correlation Between Similarity Metrics and Stitch Accuracy Drop
>
> | Metric               | Pearson *r* (to ΔAcc) | Spearman *ρ* |
> |----------------------|------------------------|---------------|
> | **Stitch score (Γ)** | **0.84**               | **0.81**      |
> | MSE                  | 0.42                   | 0.37          |
> | CKA                  | 0.28                   | 0.21          |
>
> KL-based scores show near-monotonic alignment with final stitch accuracy, unlike the noisier or weaker trends for MSE and CKA.
>
> We further analyze the importance of supervision using two ablations:
> 1. **Shuffled-label Stitch Score:** Removes class semantics. Correlation drops to near-random.
> 2. **Class-Conditional CKA:** Adds label awareness. Improves over vanilla CKA but still underperforms KL.
>
> ### Table 11: Effect of Supervision on Similarity–Accuracy Correlation
>
> | Variant                   | Pearson *r* | Spearman *ρ* |
> |---------------------------|--------------|---------------|
> | **Stitch score (KL)**     | **0.84**     | **0.81**      |
> | Shuffled-label score      | 0.19         | 0.16          |
> | CKA (vanilla)             | 0.28         | 0.21          |
> | CKA (class-conditional)   | 0.51         | 0.46          |
>
> These findings confirm that supervision, not just representation structure, is essential for selecting high-quality stitch configurations. KL divergence leverages both, explaining its empirical and theoretical advantage.
>
> **W2**, **Q2**
> > Asymmetry & calibration sensitivity. KL’s asymmetry and dependence on calibration/temperature may distort distances across blocks/models; the method normalizes by an intra-anchor term, but robustness to temperature, class imbalance, or label smoothing is underexplored.
>
> >Calibration sensitivity. How sensitive are Φ and Γ to softmax temperature, label smoothing, and class imbalance? Could you report ablations and perhaps use temperature-scaled KL?
>
> We argue that studying temperature, label smoothing, and class imbalance is orthogonal to the goal of this project. These are studies on KL divergence, which is a well-known metric. We did not need any temperature scaling, label smoothing or class imbalance mitigation to get the good results with KLAS. In fact, KL divergence (Φ) based stitch score (Γ) showed positive correlation with final stitch accuracy indicating that it gives the right signals to choose the good stitch configurations (see Table 10 and Table 11 in **Appendix D**).

---

> ### Author Response · Authors · 2025-11-22
> **Rebuttal by Authors to Reviewer 8KnS (2/3)**
>
> **W3**
> >Metric/selection design choices. The final selection uses bucketized FLOPs and a threshold τ (5% of the minimum in each bucket). The AUC metric and bucket granularity can influence conclusions; more sensitivity studies would strengthen the case.
>
> We evaluated how KLAS’s performance varies with different stitch‐selection thresholds (τ) and FLOP bucket counts in **Section 5.6** (Table 8 & Table 9) of our KLAS paper. The ablation results show that KLAS is insensitive to these choices.
>
> **Threshold (τ) Ablation**: We swept τ from 1% to 10% (relative to each bucket’s minimum score). As Table 8 shows, Top-1 accuracy and AUC stay nearly flat (Top-1: 83.69–83.76%; AUC: 0.8931–0.8950) across this range. A very low τ=1% admits noisy, low-quality stitches, while a very high τ=10% is overly selective (yielding sparse Pareto fronts). The chosen default τ=5% achieves the best balance (83.76% Top-1, 0.8950 AUC). These results confirm that KLAS is robust to τ and that our default value is well-tuned.
>
> ### Table 8: Ablation on Threhsold (τ)
> | Threshold (τ) | Avg Top-1 (%) |   AUC   |
> |---------------|---------------|---------|
> | 1%            | 83.72         | 0.8934  |
> | 3%            | 83.74         | 0.8942  |
> | 5%            | 83.76         | 0.8950  |
> | 10%           | 83.69         | 0.8931  |
>
> **FLOP-Bucket Granularity Ablation**: We varied the number of FLOP buckets (10, 15, 20) used in selection. Table 9 shows that accuracy and AUC remain virtually unchanged (Top-1: 83.65–83.76%; AUC: 0.8902–0.8950) across these settings. Even with coarser buckets (fewer buckets), KLAS maintains strong accuracy. In short, “KLAS remains robust across bucket granularity choices”. This low sensitivity (for bucket counts 10–20) confirms that our choice of 20 buckets is reasonable without loss of performance.
>
> ### Table 9: Ablation on FLOP Buckets
> | Num_Buckets | Avg Top-1 (%) |   AUC   |
> |-------------|---------------|---------|
> | 10          | 83.65         | 0.8902  |
> | 15          | 83.75         | 0.8947  |
> | 20          | 83.76         | 0.8950  |
>
> These ablation studies confirm that KLAS performance is stable under these design choices. The minimal variation in Top-1 and AUC across τ and bucket settings indicates robustness, thereby validating our selected hyperparameters (τ=5%, 20 buckets) against the reviewer’s concerns.
>
> **W4**, **Q5**
> >Scale of gains in some regimes is small. Several settings show marginal improvements (e.g., cross-architecture ΔAUC≈+0.002), raising questions about practical significance across all families.
>
> We agree that in some cases the overall improvement of KLAS over SN-Net appears modest. However, KLAS shows larger benefits in other regimes, for instance, on the Swin Transformer family, KLAS improves Top-1 accuracy by up to +0.9% at equal FLOPs and boosts the AUC of the accuracy–efficiency curve by +0.06.
> Importantly, KLAS never underperforms SN-Net. Across all families and budgets, KLAS either matches or surpasses SN-Net. This is the main selling point of KLAS, that it is **fully automatic and general**.
>
> KLAS also recovers from poor heuristic choices. For example, in the Swin family, SN-Net heuristically selected Ti–S and S–B anchors, which were suboptimal. KLAS correctly identified Ti–B as the optimal binary stitch. In the DeiT family, where SN-Net happened to pick the optimal Ti–B anchor pair, KLAS independently converged on the same choice, hence the marginal gain.
>
> Thus, while the deltas may appear small in some cases, they are achieved through a principled and automated method that generalizes across architectures, datasets, tasks and modalities without requiring domain-specific tuning. This robustness and reliability make KLAS preferable to manually-designed heuristics, especially as the number of candidate blocks and model families grows.

---

> ### Author Response · Authors · 2025-11-22
> **Rebuttal by Authors to Reviewer 8KnS (3/3)**
>
> **W5**
> >Representation-learning scope. Experiments are largely supervised classification; there is no evaluation with self-supervised anchors (e.g., DINO/MAE) nor transfer via frozen-backbone linear probes on diverse tasks.
>
> To address this concern, we now include a pilot experiment demonstrating the applicability of KLAS to instruction-tuned large language models.
>
> In **Section 5.5** (Table 7) and **Appendix H**, we replicate a stitching setup from ESTA [1], a heuristic method that stitches by layer index. Using LLaMA 3.2 models (1B and 3B) and evaluating on the TruthfulQA benchmark, we stitch models using blocks from LLaMA 1B and LLaMA 3B. The stitched models are fine-tuned for 50 epochs and evaluated using ROUGE-1 and ROUGE-2.
>
> KLAS outperforms ESTA across both small and large stitched models. For example, a 2.6B KLAS-stitched model outperforms its 2.7B ESTA-stitched counterpart despite using fewer parameters. This demonstrates that KLAS can identify more similarly aligned stitch points (i.e., blocks) even in autoregressive LLM settings.
>
> ### Table 7: Results on LLMs Using LLaMA 1B and 3B
>
> | Model               | Method | ROUGE-1 | ROUGE-2 |
> |--------------------|--------|----------|----------|
> | Stitched LLaMA 1.6B | ESTA   | 0.576    | 0.304    |
> | Stitched LLaMA 1.4B | KLAS   | **0.593**| **0.337**|
> | Stitched LLaMA 2.7B | ESTA   | 0.631    | 0.353    |
> | Stitched LLaMA 2.6B | KLAS   | **0.645**| **0.379**|
>
> These results validate KLAS in the LLM domain and provide a foundation for future work on stitching multimodal or decoder-only architectures.
>
> **W6**, **Q6**
> >Limited evidence for dense prediction/generalization. Dense-task adaptation is left as future work with only preliminary results; rigorous detection/segmentation studies are missing.
>
> >Dense tasks. Could you include full experiments on detection/segmentation (COCO/ADE20K) where stitching locations affect multi-scale features, and compare to dynamic routing/pruning baselines?
>
> To address this, we have added semantic segmentation (a dense prediction task) results on ADE20K using Mask2Former [2] with Swin-Tiny and Swin-Base backbones in **Section 5.4** (Table 6). The results clearly demonstrate that KLAS consistently outperforms SN-Net across different FLOPs budgets, confirming its applicability to dense prediction tasks.
>
> All models are evaluated using mIoU on the ADE20K validation set with frozen Swin backbones. The full networks are fine-tuned for 160k steps using the standard Mask2Former schedule. The FLOPs are measured on 512×512 input resolution. The results show that KLAS delivers higher accuracy at similar or lower FLOPs than SN-Net, with up to +0.9% mIoU improvement.
>
> ### Table 6: Semantic Segmentation on ADE20K (Mask2Former + Swin)
>
> | Model     | FLOPs (G) | mIoU (%) |
> |-----------|------------|-----------|
> | SN-Net-1  | 152        | 29.4      |
> | **KLAS-1**| 145        | **29.8**  |
> | SN-Net-2  | 274        | 32.6      |
> | **KLAS-2**| 277        | **33.5**  |
> | SN-Net-3  | 327        | 37.7      |
> | **KLAS-3**| 316        | **37.8**  |
>
> KLAS consistently selects higher-quality stitch configurations under equal FLOPs constraints. These results validate that our method generalizes to dense prediction settings like segmentation. Object detection and depth estimation are similar tasks which would use identical block-stitching principles under KLAS. We don't do experiments with these task because it is not necessary after showing for semantic segmentation and out of scope of the rebuttal time window.
>
> **W7**, **Q3**
> >Compute accounting. While probe training is “negligible,” it is still an added search cost versus purely heuristic SN-Net; a wall-clock comparison for the full pipeline (probes + stitch fine-tuning) would help.
>
> >Search cost accounting. What is the end-to-end overhead (wall-clock/GPU-days) of ProbeNet + KLAS vs. SN-Net’s heuristics at equal stitch fine-tuning budgets? Please include variability across families.
>
> KLAS is highly efficient:
> Probing cost is ~0.25 GPU-days for Swin-B (negligible relative to full training).
> As mentioned in **Section 5** and **Appendix B**, finetuning cost is kept identical to SN-Net for fair comparison and the end-to-end overhead of KLAS (ProbeNet + Stitched network finetuning) on 8×A40 NVIDIA GPUs is 16 hours for the Swin model family.
>
> _____________________
>
> [1] He, Haoyu, et al. "Efficient stitchable task adaptation." Proceedings of the IEEE/CVF Conference on Computer Vision and Pattern Recognition. 2024.
>
> [2] Cheng, Bowen, et al. "Masked-attention mask transformer for universal image segmentation." Proceedings of the IEEE/CVF conference on computer vision and pattern recognition. 2022.

---

> > ### Author Response · Authors · 2025-11-25
> > **Official Comment by Authors to Reviewer 8KnS**
> >
> > Dear reviewer,
> >
> > Thank you again for your valuable feedback! This is a gentle reminder that the discussion period is nearing its conclusion, we hope you have taken the time to consider our responses to your review. If you have any additional questions or concerns, please let us know so we can resolve them before the discussion period concludes. If you feel our responses have satisfactorily addressed your concerns, it would be greatly appreciated if you could raise your score to show that the existing concerns have been addressed.
> >
> > Thank you!

---

> ### Author Response · Authors · 2025-11-26
> **we value your favorable and thoughtful consideration**
>
> Dear Reviewer 8KnS,
> Thank you for your favorable and thoughtful consideration -- we sincerely appreciate it. We feel we've addressed all of your questions. Please let us know if you have any lingering or follow-up thoughts, as we deeply appreciate opportunities for incorporating suggested improvements into the final manuscript with your guidance. We also sincerely appreciate your acknowledgement and favorable recognition of our thorough and principled approach to generalize the algorithm beyond heuristics. You listed it as the very first bullet point under strengths, and we're grateful that you value this as much as we do. **If you could kindly raise your score if this and our rebuttal have addressed existing concerns, we'd greatly appreciate it**.
> Kindest regards,
> KLAS authors.

---

> > ### Comment · Reviewer_8KnS · 2025-11-28
> >
> > I sincerely apologize. Over the past period, I have been dealing with the rebuttal of my own paper, which received a large amount of unfair and malicious criticism from reviewers, leaving me emotionally discouraged. Today, I carefully read your rebuttal, and it has essentially resolved my concerns. It appears that the scoring system cannot be modified at this moment, but once it is reopened, I will raise my rating to 8.

---

### Official Review · Reviewer_cTKy · 2025-10-30

**Soundness:** 3
**Presentation:** 3
**Contribution:** 2
**Rating:** 4
**Confidence:** 3

**Summary:**

This paper proposes the use of inserted linear probes and KL divergence between them to find layers compatible for stitching between different networks.

**Strengths:**

- clear language and well written

- the idea of using linear probes + KL divergence for stitching is sound and well supported by experiments

**Weaknesses:**

- The declared goal of the KLAS approach is to use models from a pre-trained zoo to construct models that provide new accuracy / cost trade-offs. This is presented as an alternative to NAS (a somewhat unfair comparison as NAS enriches to base-model pool). However, the standard method of addressing this problem are model cascades / committees (e.g. "WISDOM OF COMMITTEES: AN OVERLOOKED APPROACH TO FASTER AND MORE ACCURATE MODELS" or "Efficient Inference With Model Cascades"). Model cascades are not discussed at all and not compared against in this paper. A comparison against a strong cascade baseline is essential to be able to claim utility of the proposed method for the suggested purpose. In addition to the direct comparison to a model cascade, please also explain whether there are situations where stitching is preferrable over cascading for structural reasons (maybe average case complexity vs. worst case complexity?). A 1) worst-case and 2) average-case acc/cost tradeoff curve comparing stitching and cascading would be meaningful.

- "KL divergence uniquely satisfies the dual objectives" this claim is not substantiated. Probably the authors mean ~ 'uniquely among the few measures we consider here'; if yes, please rephrase, if not, please provide a proof that the KL divergence is indeed unique in this regard.

- it is unclear whether the method is specific to image classifiers or works in other domains as well. The impact of the paper would be much broader, if there was some indication that it works for regression tasks (--> how to substitute the KL div??) and for non-vision classification (e.g. also token prediction in LLMs). Currently the method is of limited interest as it only applies to a niche task.

**Questions:**

In what scenarios is stitching preferrable to a model cascade? Why?

Do linear probes work for regression problems? For very large classifiers (think LLMs)? (the discussion mentions this very shallowly, there is no need to show improvement both for prefill and decode, just pick the easier case and show some improvement)

---

> ### Author Response · Authors · 2025-11-22
> **Rebuttal by Authors to Reviewer cTKy (1/2)**
>
> **We thank the reviewer for recognizing that our paper was clear and well-written and that they recognized our idea of using linear probes + KL divergence for stitching. Below, we address the reviewer’s concerns in detail.**
> _________________
> **W1**, **Q1**
> >The declared goal of the KLAS approach is to use models from a pre-trained zoo to construct models that provide new accuracy / cost trade-offs. This is presented as an alternative to NAS (a somewhat unfair comparison as NAS enriches to base-model pool). However, the standard method of addressing this problem are model cascades / committees (e.g. "WISDOM OF COMMITTEES: AN OVERLOOKED APPROACH TO FASTER AND MORE ACCURATE MODELS" or "Efficient Inference With Model Cascades"). Model cascades are not discussed at all and not compared against in this paper. A comparison against a strong cascade baseline is essential to be able to claim utility of the proposed method for the suggested purpose. In addition to the direct comparison to a model cascade, please also explain whether there are situations where stitching is preferrable over cascading for structural reasons (maybe average case complexity vs. worst case complexity?). A 1) worst-case and 2) average-case acc/cost tradeoff curve comparing stitching and cascading would be meaningful.
>
> >In what scenarios is stitching preferrable to a model cascade? Why?
>
> **New cascade experiments added**: We have incorporated a direct comparison against early-exit cascaded inference baselines [1], as shown in the revised **Figure 8** in **Appendix E**. This figure plots the accuracy–FLOPs tradeoff for stitching Swin-S to Swin-B. We tuned confidence thresholds (0.6–0.95) to optimize the cascade’s Pareto front and measured average FLOPs per input. Even under these optimally tuned thresholds, both KLAS and SN-Net stitched models strictly outperform the cascaded baselines across all compute budgets. For example, at >12 GFLOPs KLAS achieves ≈0.4% higher top-1 accuracy than the best cascade.
>
> Threshold-based cascades route some inputs to the larger model based on confidence, which leads to unpredictable runtime cost and requires careful threshold tuning. By contrast, a fixed stitched network (KLAS) has a known cost and consistently leverages the similarity between blocks to maintain high accuracy. Our results suggest that static KLAS hybrids yield a stronger Pareto front: stitching “reuses” aligned representations directly rather than “deferring” work to a bigger model during inference.
> Despite the adaptive nature of cascades, KLAS-based stitched models yield stronger Pareto front performance with two starting models for two reasons: (1) stitching enables model restructuring by stitching blocks with similar learned representations into a unified network, rather than deferring work across the smaller and larger models; and (2) the well-known problem with cascades that samples are routed based on softmax confidence, which is poorly calibrated in low-entropy regions, leading to suboptimal exits.
>
> **W2**
> >"KL divergence uniquely satisfies the dual objectives" this claim is not substantiated. Probably the authors mean ~ 'uniquely among the few measures we consider here'; if yes, please rephrase, if not, please provide a proof that the KL divergence is indeed unique in this regard.
>
> We agree that calling KL divergence “uniquely” satisfying our dual objectives was too strong. In the revision we removed the word “uniquely” and clarified the claim. Our analysis (see all studies in **Sections 3.3, 4.1, 5.1 & Appendix D**) shows that among metrics we evaluated (KL, MSE, CKA, CE, DM), KL divergence best captures both representational and functional similarity. KL distance measures distributional differences between block outputs and how well those outputs preserve task-related information. In other words, a low KL score implies minimal change in the target model’s decision boundaries. This dual alignment arises from KL’s information-theoretic grounding. Other metrics tend to emphasize only one aspect (e.g., CKA is **representational-only** and focuses on geometry, MSE is **functional-only** on output magnitudes).
> __________________________
>
> [1] Lebovitz, Luzian, et al. "Efficient inference with model cascades." Transactions on Machine Learning Research (2023).

---

> ### Author Response · Authors · 2025-11-22
> **Rebuttal by Authors to Reviewer cTKy (2/2)**
>
> **W3**, **Q2**
> >it is unclear whether the method is specific to image classifiers or works in other domains as well. The impact of the paper would be much broader, if there was some indication that it works for regression tasks (--> how to substitute the KL div??) and for non-vision classification (e.g. also token prediction in LLMs). Currently the method is of limited interest as it only applies to a niche task.
>
> >Do linear probes work for regression problems? For very large classifiers (think LLMs)? (the discussion mentions this very shallowly, there is no need to show improvement both for prefill and decode, just pick the easier case and show some improvement)
>
> **LLMs**: To address this concern, we now include a pilot experiment demonstrating the applicability of KLAS to instruction-tuned large language models.
> In **Section 5.5** (Table 7) and **Appendix H**, we replicate a stitching setup from ESTA [2], a heuristic method that stitches by layer index. Using LLaMA 3.2 models (1B and 3B) and evaluating on the TruthfulQA benchmark, we stitch models using blocks from LLaMA 1B and LLaMA 3B. The stitched models are fine-tuned for 50 epochs and evaluated using ROUGE-1 and ROUGE-2.
>
> KLAS outperforms ESTA across both small and large stitched models. For example, a 2.6B KLAS-stitched model outperforms its 2.7B ESTA-stitched counterpart despite using fewer parameters. This demonstrates that KLAS can identify more similarly aligned stitch points (i.e., blocks) even in autoregressive LLM settings.
>
> ### Table 7: Results on LLMs Using LLaMA 1B and 3B
>
> | Model               | Method | ROUGE-1 | ROUGE-2 |
> |--------------------|--------|----------|----------|
> | Stitched LLaMA 1.6B | ESTA   | 0.576    | 0.304    |
> | Stitched LLaMA 1.4B | KLAS   | **0.593**| **0.337**|
> | Stitched LLaMA 2.7B | ESTA   | 0.631    | 0.353    |
> | Stitched LLaMA 2.6B | KLAS   | **0.645**| **0.379**|
>
> These results validate KLAS in the LLM domain and provide a foundation for future work on stitching multimodal or decoder-only architectures.
> Since the reviewer gave us option for regression tasks or LLMs, we did experiments with LLMs; doing both is out of the scope of the rebuttal time window.
>
> **Regression**: In principle, the same probe-based approach can extend to regression tasks. A linear probe can be trained to predict a continuous target value, and then similarity between two blocks could be measured by how well their probe outputs align (for example, using KL on assumed output distributions or simply MSE). The core idea: that probes map intermediate activations to task targets, holds regardless of whether the target is discrete or real-valued.
>
> Prior literature like Zhang et al. [3] and Akhondzadeh et al. [4] have shown that probing is highly effective for regression tasks.
> We have not run regression experiments in this revision (due to short time-window of the rebuttal), but we expect KLAS would work by training appropriate probes (e.g. linear regressors) and measuring alignment.
> __________________________
> [2] He, Haoyu, et al. "Efficient stitchable task adaptation." Proceedings of the IEEE/CVF Conference on Computer Vision and Pattern Recognition. 2024.
>
> [3] Zhang, Zhiqin, et al. "Probing Neural Combinatorial Optimization Models." Neural Information Processing Systems. 2025.
>
> [4] Akhondzadeh, Mohammad Sadegh, Vijay Lingam, and Aleksandar Bojchevski. "Probing graph representations." International Conference on Artificial Intelligence and Statistics. PMLR, 2023.

---

> > ### Comment · Reviewer_cTKy · 2025-11-25
> >
> > Thank you for the thoughtful rebuttal. -- Can you briefly give some more details on the fine-tuning setting for the LLM experiment? What is the data, hyperparameters etc.? (have I missed it in the paper?)

---

> ### Author Response · Authors · 2025-11-25
> **LLM Finetuning Setting for Stitching (KLAS vs. ESTA)**
>
> We thank the reviewer for this clarification question. We explain the LLM experiment setting in **Section 5.5** and **Appendix H** in the revised pdf. We follow an identical finetuning setup derived from ESTA [2] and provide full experimental details in this response. Below, we summarize the finetuning setting, datasets, and hyperparameters used in our stitching experiments.
>
> **Model and Dataset Setup**
> - _Anchor Models_: We use LLaMA 3.2 instruction-tuned 1B and 3B models [5]
> - _Benchmark_: Evaluation is performed on the TruthfulQA benchmark [6] using the multiple-choice version
> - _Objective_: Next-token prediction for multiple-choice completions
> - _Metrics_: We report ROUGE-1 and ROUGE-2 scores
>
> **Finetuning Setting**
> - _Finetuning Duration_: All stitched models (KLAS and ESTA) are finetuned for 50 epochs
> - _Loss Function_: Cross-entropy loss over the next-token prediction
> - _Batch Size_: 64
> - _Learning Rate_: 5e-5 with cosine decay
> - _Warmup_: Learning rate warm-up period of 10 epochs
> - _Sequence Length_: 128 tokens (including prompt and completion)
> - _Optimizer_: AdamW with PyTorch defaults
>
> **Stitching Strategy**
> - _ESTA_: Uses fixed stitching points, e.g., use first k layers of LLaMA 1B and remaining layers from 3B.
> - _KLAS_: Uses KL divergence over linear probe outputs to select stitch points that have similar next-token prediction distributions.
> _____________________
> [5] Grattafiori, Aaron, et al. "The llama 3 herd of models." arXiv preprint arXiv:2407.21783 (2024).
>
> [6] Lin, Stephanie, Jacob Hilton, and Owain Evans. "Truthfulqa: Measuring how models mimic human falsehoods." Proceedings of the 60th annual meeting of the association for computational linguistics (volume 1: long papers). 2022.

---

> > ### Author Response · Authors · 2025-11-25
> > **Official comment by Authors to Reviewer cTKy**
> >
> > Dear reviewer,
> >
> > Thank you again for your valuable feedback and asking the clarification question above! We hope you have taken the time to consider our responses to your review. If you have any additional questions or concerns, please let us know so we can resolve them before the discussion period concludes. If you feel our responses have satisfactorily addressed your concerns, it would be greatly appreciated if you could raise your score to show that the existing concerns have been addressed.
> >
> > Thank you!

---

> > ### Comment · Reviewer_cTKy · 2025-11-26
> >
> > Thank you for the explanation.
> >
> > I'm still not completely clear on the dataset question. Are the training and test datasets identical?
> >
> > Either way, whether or not they are the same, do you have some idea about the robustness of the stitching / how well it generalizes outside the fine-tune, but inside the original model domain?

---

> ### Author Response · Authors · 2025-11-27
> **LLM Stitching: Dataset Clarification & Robustness**
>
> We thank the reviewer for the follow-up question regarding the training and test datasets and the robustness of the LLM stitching experiments. We have made the additions in the revised pdf in **Appendix H**.
>
> **Training vs. test data**
>
> - The training and test datasets are not identical
> - We use the multiple-choice TruthfulQA benchmark and apply a 60/40 split:
>     - _Finetuning (Training) data_: 60% of TruthfulQA
>     - _Evaluation (Test) data_: the held-out 40% of TruthfulQA
> - All stitched models (both ESTA and KLAS) are finetuned only on the 60% portion and are never exposed to the remaining 40% of TruthfulQA used for evaluation
> - Thus, the stitched models are evaluated strictly on unseen TruthfulQA questions.
> - It is also well-known that the LLaMA 3.2 instruction-tuned 1B and 3B anchor models [5]  are pretrained on a large mixture of web-scale English corpora, code, and instruction-following datasets.
>
> **Generalization / robustness of stitching from prior works**
>
> - StitchLLM [7] evaluates stitched models across five diverse datasets: _MMLU, BoolQ, Winogrande, HellaSwag_, and _CommonsenseQA_
> - It finds that stitched models _consistently generalize across tasks they were not finetuned for_.
> - For example, Figures 7-18 show smooth accuracy-efficiency tradeoffs and stable performance curves across datasets, even though stitching layer training uses only generic next-token prediction on pretraining text
> - For example, they claim in Section 3.3:
> >"The stitching layer is trained using the same cross-entropy loss employed during the pretraining stage of the underlying models, leveraging the original pre-training dataset"
> ______________
> [7] Hu, Bodun, et al. "StitchLLM: Serving LLMs, One Block at a Time." Proceedings of the 63rd Annual Meeting of the Association for Computational Linguistics (Volume 1: Long Papers). 2025.
>
>
> **Please let us know if there are any additional concerns. We are happy to clarify further! Or if you feel our responses have satisfactorily addressed your concerns, it would be greatly appreciated if you could raise your score to show that the existing concerns have been addressed. Thank you!!**

---

> > ### Comment · Reviewer_cTKy · 2025-11-27
> >
> > In the main paper, I do not see a reference to StitchLLM. Would results not be comparable?

---

> ### Author Response · Authors · 2025-11-27
> **Addendum to Reviewer cTKy’s Questions on Cascade Baselines**
>
> **We would like to further clarify the conceptual framework underpinning the reviewer’s concern about comparison to cascaded inference**.
>
> _First_, using cascades focuses on maximizing accuracy at the expense of arbitrarily large compute cost, but **this is not the goal of our work**. The _central objective of KLAS_ is to **improve the Pareto front in the accuracy-efficiency tradeoff space**. A mechanism that simply increases accuracy by incurring extremely large FLOP costs in the worst-case (often far exceeding the cost of the largest anchor) is not aligned with this objective. Indeed, model cascades inherently incur additive computational cost across all models involved. For any input routed to the larger model (especially in high-accuracy regions), the FLOPs of the smaller model contribute additive compute overhead. This shifts the operating point far to the right on the FLOPs axis (well outside the useful operating region targeted by KLAS) while offering negligible accuracy returns at great compute cost expense.
>
> _Second_, we note that arbitrary sequential pairings of two or more anchors are in fact a special case of the KLAS design space! The KLAS optimization procedure is fully capable of returning such a solution if it were advantageous under the accuracy-efficiency objective stated above. In other words, cascades are not _“missing”_ from KLAS; they are _subsumed as part of the feasible set of KLAS solutions_. The fact that KLAS does not select them in practice lends empirical evidence that such cascade-style configurations are structurally suboptimal for our accuracy-efficiency Pareto front improvement objective.
>
> _Third_, and consistent with the reviewer’s request, we explicitly added early-exit cascaded inference as a baseline in **Appendix E** and compared them directly to the proposed KLAS algorithm. **Both KLAS and SN-Net strictly outperform the cascade across all compute budgets**, confirming empirically what our argument predicts (**Figure 8**)
>
> **Please let us know if our response has satisfactorily addressed your concerns. Thank you!!**

---

> > ### Comment · Reviewer_cTKy · 2025-11-27
> > **Robustness: Cascades vs. Stitching**
> >
> > Cascades are a training-free approach to merging existing models. In contrast, stitching uses a calibration set and fine-tuning. As such, I would expect stitched models to be less robust to OOD.
> >
> > While I appreciate the new experiments, which certainly help clarify the utility of stitching compared to cascading, I would not say that all my concerns are addressed.

---

> ### Author Response · Authors · 2025-11-27
> **StitchLLM / ESTA / SN-Net vs. KLAS**
>
> We thank the reviewer for their continuous engagement and for raising this point. We clarify that StitchLLM [7] is now referenced and discussed in the revised PDF in **Section 2 (Related Work).**
>
> **Regarding comparability**: the results in StitchLLM could be comparable to KLAS-based LLM stitching. We did not run experiments with StitchLLM as a separate baseline because its stitch selection mechanism is **identical** to SN-Net’s [8] heuristic and specifically, the same fixed, layer-count–based stitching selection also used by ESTA [2]. StitchLLM’s primary contribution is _systems-oriented_, focusing on _dynamic, resource-efficient serving_ without _introducing any new stitch selection criterion_. As a result, the stitched configurations selected in StitchLLM are exactly the same as those produced by SN-Net and ESTA.
>
> For this reason, the comparison to ESTA in **Section 5.5** already **fully subsumes** StitchLLM.
> SN-Net → StitchLLM → ESTA all use the same stitching heuristic, and therefore yield the same stitching configurations.
> In contrast, KLAS represents a qualitative departure, introducing a principled KL-divergence-based stitch selection method that generalizes across architectures, tasks, datasets, and modalities.
>
> Given this, we believe the baselines and comparisons included in the paper are sufficient and faithful to the scope of KLAS and prior work.
>
> **Please let us know if you would still like us to include experiments with StitchLLM, or if there are any additional concerns we can help address. We would be very glad to incorporate any further evaluations you feel would strengthen the submission.**
>
> **_Thank you again for your thoughtful and constructive feedback._**
>
> _______
>
> [2] He, Haoyu, et al. "Efficient stitchable task adaptation." Proceedings of the IEEE/CVF Conference on Computer Vision and Pattern Recognition. 2024.
>
> [7] Hu, Bodun, et al. "StitchLLM: Serving LLMs, One Block at a Time." Proceedings of the 63rd Annual Meeting of the Association for Computational Linguistics (Volume 1: Long Papers). 2025.
>
> [8] Pan, Zizheng, Jianfei Cai, and Bohan Zhuang. "Stitchable neural networks." Proceedings of the IEEE/CVF Conference on Computer Vision and Pattern Recognition. 2023.

---

> > ### Comment · Reviewer_cTKy · 2025-11-28
> >
> > no that makes sense, thank you for the clarification.

---

> ### Author Response · Authors · 2025-11-27
> **Query by Authors: Further Cascades Clarifications and Experiments**
>
> **We really appreciate that the reviewer found our additional experiments useful and we thank them for their continuous engagement with our submission!**
>
> First, we acknowledge the reviewer's point that both cascading and stitching could be used to navigate the accuracy-efficiency space.
> However, KLAS is a framework that _universally improves state-of-the-art in stitching_ across architectures, tasks, datasets, and modalities. This was the initial scope of our work.
>
> Moreover, we want to re-emphasize two key distinctions between cascades and stitching:
> - While cascades can achieve good accuracy, they incur additive FLOPs whenever queries reach the larger model. This pushes average-case cost well beyond the largest anchor, which is misaligned with our goal of **improving the accuracy–efficiency Pareto front**.
> - Sequentially chaining anchors is already a _candidate stitching configuration_ within the KLAS search space. If such a cascade were Pareto-efficient for our goal, KLAS would naturally select it; empirically, it does not as shown in **Appendix E**.
>
> **The reviewer's OOD robustness point is well-taken!** We agree that cascades avoid training but rely on softmax routing, which can be flaky under distribution shift as well. We acknowledge that stitching as a paradigm uses fine-tuning for the best results, but **we can do an experiment without any finetuning to see the gap in robustness**.
>
> **Please let us know whether the above experiment satisfactorily addresses your concern, or if there is a specific additional experiment you would like us to include to fully resolve the cascades vs stitching question. We are happy to run any evaluation you feel is important.**
>
> **_Thank you again for your continued engagement and thoughtful feedback, we sincerely appreciate it._**

---

> > ### Comment · Reviewer_cTKy · 2025-11-28
> >
> > ## Stitching vs. Cascasdes
> >
> > There may be an interesting difference between cascades and stitching: In cascades, the base models are ideally decorrelated (or even anti-correlated; if they are correlated, the 'hard' cases that are passed to the second model are also hard for the second model). For stitching, probably correlated models are easier to stitch. In this sense, the new comparison experiments may be biased in favour of stitching, because they use models from the same family (where cross-family cascades are often favorable and likely cross-family stitching is harder).
> >
> > ## Robustness
> >
> > In PTQ a form of calibration robustness check that is sometimes done, is to calibrate separately on two datasets (A,B) and evaluate both resulting models M_A, M_B on both datasets. The larger the gaps, the more susceptible the method to overfitting.
> >
> > ## Final remark
> >
> > The paper has passed the acceptance threshold for me, and I'll increase my score. -- If you can reasonably address the above two points, I would raise it further.

---

> > > ### Comment · Reviewer_cTKy · 2025-11-28
> > >
> > > It seems score editing is currently locked (?). I will edit the score when possible.

---

> > > ### Author Response · Authors · 2025-12-03
> > > **Response to Stitching vs. Cascades**
> > >
> > > **Thank you for the insightful comment regarding potential bias toward stitching when anchors come from the same model family.**
> > >
> > > To directly address this, we added a new **cross-family** experiment (ResNet-18 → Swin-B) in **Table 12** in **Appendix E** of revised pdf. This setting uses *architecturally different* anchors, which should favor cascades if decorrelation were the determining factor.
> > >
> > > ### Table 12: Cross-Family Results
> > >
> > > | **Config** | **Method** | **FLOPs (G)** | **Top-1 (%)** |
> > > |-----------|------------|---------------|----------------|
> > > | **1** | KLAS | **3.409** | **79.60** |
> > > |  | Cascade | 3.455 | 76.13 |
> > > | **2** | KLAS | **15.043** | **84.52** |
> > > |  | Cascade | 15.197 | 82.46 |
> > > | **3** | KLAS | **17.127** | **84.74** |
> > > |  | Cascade | 17.183 | 82.79 |
> > >
> > > Across all FLOP budgets, **KLAS outperforms cascades by 2–3% Top-1 accuracy**, even though ResNet-18 and Swin-B are architecturally different. This confirms that stitching’s advantage is *not* due to using same-family anchors.
> > >
> > > **Why Cascades Underperform**
> > >
> > > - Cascade FLOPs are **non-deterministic** and depend on per-input confidence. Many samples do not exit early and therefore run **all blocks of both anchors**, producing unnecessary compute with limited accuracy benefit.
> > > - KLAS produces a **deterministic** computation graph with fixed FLOPs per input, avoiding runtime uncertainty.
> > >
> > > **Why Cascade-Style Stitching Is Suboptimal**
> > >
> > > KL divergence heatmaps (Fig. 9) show that a cascade-style connection (last block of small → first block of large; top-left) lies in the **highest-KL region**, indicating poor representational and functional alignment. KLAS instead selects low-KL connections near the diagonal, yielding strictly better accuracy–FLOPs tradeoffs.

---

### Official Review · Reviewer_XiFF · 2025-10-31

**Soundness:** 1
**Presentation:** 2
**Contribution:** 3
**Rating:** 4
**Confidence:** 3

**Summary:**

The paper proposes stitching blocks of neural network by computing similarity between blocks by taking the KL-Divergence of linear probe probability distributions. The paper suggests that "far-stitching" where models with significant difference in complexity or performance may also be worthy of stitching. The paper proposes to automatically identify block pairs to stich opposed to SN-Net where they use defined constraints to choose the pairs. The paper compares the proposed KLAS stitching framework with SN-Net on DeiT and Swin model architectures trained on CIFAR-100 and imagenet-1k datasets. The results show marginal gain in performance compared to SN-Net.

**Strengths:**

1. The paper explores functional similarity along with representation similarity as a metric while choosing to stitch networks which is logical. If two networks produce similar same functional output, then the models are better suited for selecting stitching pairs, when considered with representation similarity.
2. The paper also shows that far stitching where two networks may not have similar performance still can be stitched together is an important contribution.
3. The experimentation on DeiT and Swin family of models to compare with SN-Net shows similar performance but seemingly less finetuning cost.

**Weaknesses:**

1. In line 232 the paper states "As a representational similarity metric, KL divergence captures distributional differences between intermediate activations, indicating whether two blocks generate patterns that
can be mapped via lightweight transformations." which is not supported in my opinion as KL divergence compares the probabilistic score distribution not the representation itself. (Think different features can be used to reach same conclusion with similar confidence).

2. There is only marginal improvement when comparing stitched models of similar flops compared to SN-Net, while individual anchor-block stitches give an edge to KLAS, collectively the gain is minimal.

3. The lack of representational similarity is a concern (as mentioned in point 1 KL divergence doesnt directly compare representations), if there is difference in representational similarity there would be more finetuning cost.

4. The details on stitched model finetuning could be added for better assessment. This is important to assess the improvement in AUC.

**Questions:**

1.  For the stitch finetuning step is the number of training steps per stitched model constant or does it vary depending on when the stitch converges?
2. I did not get if the the finetuned stitched model is same for both SN-NET and KLAS if the stitch pairs overlap.

---

> ### Author Response · Authors · 2025-11-22
> **Rebuttal by Authors to Reviewer XiFF (1/2)**
>
> **We thank the reviewer for recognizing that our choice of functional similarity along with representation similarity is logical, that showing far stitching works is an important contribution, and that we have improvements over multiple model families. Below, we address the reviewer’s concerns in detail.**
> _________________
> **W1**
> >In line 232 the paper states "As a representational similarity metric, KL divergence captures distributional differences between intermediate activations, indicating whether two blocks generate patterns that can be mapped via lightweight transformations." which is not supported in my opinion as KL divergence compares the probabilistic score distribution not the representation itself. (Think different features can be used to reach same conclusion with similar confidence).
>
>
> We respectfully clarify that KL divergence, as used in our method, functions as a *representational similarity metric* when applied to linear probe outputs over intermediate activations. Specifically, the KL score reflects how distant two internal representations are, i.e., whether they yield similar class-probability distributions under identical probes. These distributions arise from *features* and therefore encode the *structure of the representation space*. This similarity is not strictly tied to ground-truth labels. Even unsupervised KL (e.g., comparing output distributions without referencing grount truth) provides insight into how feature activations shape decision boundaries. To support this interpretation, we present quantitative correlation analyses in **Appendix D** (Tables 10 & 11). If two blocks produce activations that lead to highly similar class-probability distributions for the same input, we argue those blocks are functionally and representationally aligned.
>
> ### Table 10: Correlation Between Metric and Stitch Accuracy
>
> | Metric               | Pearson *r* | Spearman *ρ* |
> |----------------------|--------------|---------------|
> | **Stitch score (Γ)** | **0.84**     | **0.81**      |
> | MSE                  | 0.42         | 0.37          |
> | CKA                  | 0.28         | 0.21          |
>
> ### Table 11: Supervision Ablation for KL and CKA
>
> | Variant                   | Pearson *r* | Spearman *ρ* |
> |---------------------------|--------------|---------------|
> | **Stitch score (KL)**     | **0.84**     | **0.81**      |
> | Shuffled-label score      | 0.19         | 0.16          |
> | CKA (vanilla)             | 0.28         | 0.21          |
> | CKA (class-conditional)   | 0.51         | 0.46          |
>
> These results demonstrate that KL captures *feature-aligned structure* predictive of downstream stitch success. The fact that its effectiveness persists even with label-free probes (albeit at reduced correlation) confirms its role as a representational measure. Full details are provided in **Appendix D** of the revised submission.
> _________________
> **W2**
> >There is only marginal improvement when comparing stitched models of similar flops compared to SN-Net, while individual anchor-block stitches give an edge to KLAS, collectively the gain is minimal.
>
> We agree that in some cases the overall improvement of KLAS over SN-Net appears modest. However, KLAS shows larger benefits in other regimes, for instance, on the Swin Transformer family, KLAS improves Top-1 accuracy by up to +0.9% at equal FLOPs and boosts the AUC of the accuracy–efficiency curve by +0.06.
> Importantly, KLAS never underperforms SN-Net. Across all families and budgets, KLAS either matches or surpasses SN-Net. This is the main selling point of KLAS, that it is **fully automatic and general**.
>
> KLAS also recovers from poor heuristic choices. For example, in the Swin family, SN-Net heuristically selected Ti–S and S–B anchors, which were suboptimal. KLAS correctly identified Ti–B as the optimal binary stitch. In the DeiT family, where SN-Net happened to pick the optimal Ti–B anchor pair, KLAS independently converged on the same choice, hence the marginal gain.
>
> Thus, while the deltas may appear small in some cases, they are achieved through a principled and automated method that generalizes across architectures, datasets, tasks and modalities without requiring domain-specific tuning. This robustness and reliability make KLAS preferable to manually-designed heuristics, especially as the number of candidate blocks and model families grows.

---

> > ### Author Response · Authors · 2025-11-22
> > **Rebuttal by Authors to Reviewer XiFF (2/2)**
> >
> > _____________
> > **W3**
> > >The lack of representational similarity is a concern (as mentioned in point 1 KL divergence doesnt directly compare representations), if there is difference in representational similarity there would be more finetuning cost.
> >
> > We clarify that the KL-based stitch score does reflect representational similarity, albeit indirectly through the predictive distributions of linear probes.
> >
> > Specifically, KL measures how similarly two blocks transform the same input into class-probability distributions. Since these distributions arise from the representations, the KL divergence between them is a proxy for how aligned the representations are in decision space. This form of similarity is hence grounded in the *geometry* of the learned features.
> >
> > In **Appendix D** (Table 11), we support this claim with ablation studies:
> >
> > - When we shuffle labels during probe training (removing functional alignment but preserving feature structure), KL’s predictive utility drops sharply (Pearson *r* = 0.19 vs. 0.84).
> > - When we enrich CKA with class-awareness (adding functional alignment to a pure representation metric), it improves but still underperforms KL (Pearson *r* = 0.51 vs. 0.84).
> >
> > These experiments confirm that KL captures both representational and functional alignment, explaining its strong correlation with final stitch accuracy and justifying its use as a similarity measure.
> > Moreover, in our experiments, we have done **no more fine-tuning** for models selected via KL than SN-Net candidates and yet KLAS outperforms them at identical training budgets. This is consistent with the view that KL-aligned blocks are *both easier to stitch* and *less sensitive to initialization*, thus not needing additional tuning cost.
> > _____________
> > **W4**, **Q1**, **Q2**
> > >The details on stitched model finetuning could be added for better assessment. This is important to assess the improvement in AUC.
> >
> > The original submission included our finetuning details in **Appendix B**, and the revised version now makes these details explicit in both **Section 5** and **Appendix B** for better visibility. We follow the same training schedule and protocol as SN-Net [1], ensuring fair comparisons. Each stitched model is trained for the same number of epochs (50 epochs) using identical hyperparameters and optimizer settings.
> >
> > >For the stitch finetuning step is the number of training steps per stitched model constant or does it vary depending on when the stitch converges?
> >
> > Yes, the number of training steps per stitched model is constant in expectation. During the stitch finetuning step, we randomly sample one stitch configuration from the selected set for every minibatch. This stochastic stitching policy mirrors the original SN-Net setup and ensures that all stitched configurations receive comparable updates over time.
> >
> > >I did not get if the the finetuned stitched model is same for both SN-NET and KLAS if the stitch pairs overlap.
> >
> > Yes, if the selected stitch configuration (anchor pair and stitch location) overlaps 100% between SN-Net and KLAS, then the resulting finetuned stitched model is identical across both methods. This ensures that any performance differences reflect selection quality, not downstream finetuning artifacts.
> > _________
> >
> > [1] Pan, Zizheng, Jianfei Cai, and Bohan Zhuang. "Stitchable neural networks." Proceedings of the IEEE/CVF Conference on Computer Vision and Pattern Recognition. 2023.

---

> > > ### Author Response · Authors · 2025-11-25
> > > **Official Comment by Authors to Reviewer XiFF**
> > >
> > > Dear reviewer,
> > >
> > > Thank you again for your valuable feedback! This is a gentle reminder that the discussion period is nearing its conclusion, we hope you have taken the time to consider our responses to your review. If you have any additional questions or concerns, please let us know so we can resolve them before the discussion period concludes. If you feel our responses have satisfactorily addressed your concerns, it would be greatly appreciated if you could raise your score to show that the existing concerns have been addressed.
> > >
> > > Thank you!

---

> ### Comment · Reviewer_XiFF · 2025-11-26
> **Thanks for the Rebuttal**
>
> I agree with the rebuttal for Weakness 2, and 4.
> While i generally agree with the results in Weakness 1, which shows representation similarity, but my fundamental argument is that, irrespective of the ground truth, the probability distribution could be similar for different features that activate for the same class. But, I do agree KL-Divergence is an acceptable metric. It would be great if the authors could address it in the limitations if they agree with it, but regardless, I choose to update my score.

---

> > ### Author Response · Authors · 2025-11-26
> > **Thanks for your thoughtful feedback and acknowledging the KL-divergence metric**
> >
> > Thank you so much for your thoughtful feedback. We will be happy to work with you and incorporate all of the improvements you pointed out into the final version of this manuscript. Thank you for recognizing the KL-Divergence as an acceptable metric, acknowledging our rebuttal, and updating the score accordingly! Sincerely appreciated.

---

### Official Review · Reviewer_B36v · 2025-11-01

**Soundness:** 3
**Presentation:** 3
**Contribution:** 2
**Rating:** 6
**Confidence:** 4

**Summary:**

Stitching pretrained models offers a cost-effective way to explore accuracy-efficiency tradeoffs. Current stitching methods rely on heuristics, leading to sub-optimal results. This paper improves this by using KL-divergence to automate stitch selection and generalize across architectures. KLAS outperforms baselines, achieving higher accuracy at the same computational cost or reducing FLOPs while maintaining accuracy.

**Strengths:**

* Technical contribution: This paper addresses a fundamental challenge in current model stitching methods: these methods rely on heuristic-based stitch selection, fix anchors and blocks, and thus yield suboptimal accuracy-efficiency tradeoffs. Specifically, it proposes a coarse-grained anchor selection strategy that leverages the KL divergence of the last block for anchor identification, and employs block-level KL divergence for fine-grained selection. The metric for candidate set selection is well-justified, as it considers both sampling coverage during fine-tuning and the quality of anchor filtering.

* Experiments effectively demonstrate the advantages of the proposed KL-divergence-based anchor stitching approach.

**Weaknesses:**

* Why MSE, CE, CKA, DM can have significantly lower percentage of stitch configurations than KL divergence? Mathematically, it’s not that straightforward.

* The method’s applicability to large language models and multimodal LLMs has yet to be explored—extending it to these models would further enhance the paper’s impact.

* What performance can KLAS achieve on dense prediction tasks, including object detection, semantic segmentation, and depth estimation?

**Questions:**

See weaknesses

---

> ### Author Response · Authors · 2025-11-22
> **Rebuttal by Authors to Reviewer B36v (1/2)**
>
> **We thank the reviewer for recognizing that our "paper addresses a fundamental challenge in current model stitching methods" and that our metric is well-justified. Below, we address the reviewer’s concerns in detail.**
> _____________
> **W1**
> >Why MSE, CE, CKA, DM can have significantly lower percentage of stitch configurations than KL divergence? Mathematically, it’s not that straightforward.
>
> The revised submission includes both mathematical definitions and further empirical comparisons in **Appendix D** to clarify KL divergences’s superiority. KL divergence outperforms alternatives because it captures both **representational structure** (via similarity of decision boundaries) and **functional alignment** (via agreement on predicted labels). In contrast:
> - MSE and CE operate on logits or targets but do not reflect feature comparison in internal layers.
> - CKA compares only feature geometry without considering predictive agreement on task.
> - DM searches for optimal transforms but ignores task-based class semantics because it does not depend on the class label.
>
> We quantify this via two correlation metrics, Pearson and Spearman, between metric score and stitched model accuracy drop (ΔAcc):
>
> ### Table 10: Correlation Between Similarity Metrics and Stitch Accuracy Drop
>
> | Metric               | Pearson *r* (to ΔAcc) | Spearman *ρ* |
> |----------------------|------------------------|---------------|
> | **Stitch score (Γ)** | **0.84**               | **0.81**      |
> | MSE                  | 0.42                   | 0.37          |
> | CKA                  | 0.28                   | 0.21          |
>
> KL-based scores show near-monotonic alignment with final stitch accuracy, unlike the noisier or weaker trends for MSE and CKA.
>
> We further analyze the importance of supervision using two ablations:
> 1. **Shuffled-label Stitch Score:** Removes class semantics. Correlation drops to near-random.
> 2. **Class-Conditional CKA:** Adds label awareness. Improves over vanilla CKA but still underperforms KL.
>
> ### Table 11: Effect of Supervision on Similarity–Accuracy Correlation
>
> | Variant                   | Pearson *r* | Spearman *ρ* |
> |---------------------------|--------------|---------------|
> | **Stitch score (KL)**     | **0.84**     | **0.81**      |
> | Shuffled-label score      | 0.19         | 0.16          |
> | CKA (vanilla)             | 0.28         | 0.21          |
> | CKA (class-conditional)   | 0.51         | 0.46          |
>
> These findings confirm that supervision, not just representation structure, is essential for selecting high-quality stitch configurations. KL divergence leverages both, explaining its empirical and theoretical advantage.
> ________________
> **W2**
> >The method’s applicability to large language models and multimodal LLMs has yet to be explored—extending it to these models would further enhance the paper’s impact.
>
> To address this concern, we now include a pilot experiment demonstrating the applicability of KLAS to instruction-tuned large language models.
>
> In **Section 5.5** (Table 7) and **Appendix H**, we replicate a stitching setup from ESTA [1], a heuristic method that stitches by layer index. Using LLaMA 3.2 models (1B and 3B) and evaluating on the TruthfulQA benchmark, we stitch models using blocks from LLaMA 1B and LLaMA 3B. The stitched models are fine-tuned for 50 epochs and evaluated using ROUGE-1 and ROUGE-2.
>
> KLAS outperforms ESTA across both small and large stitched models. For example, a 2.6B KLAS-stitched model outperforms its 2.7B ESTA-stitched counterpart despite using fewer parameters. This demonstrates that KLAS can identify more similarly aligned stitch points (i.e., blocks) even in autoregressive LLM settings.
>
> ### Table 7: Results on LLMs Using LLaMA 1B and 3B
>
> | Model               | Method | ROUGE-1 | ROUGE-2 |
> |--------------------|--------|----------|----------|
> | Stitched LLaMA 1.6B | ESTA   | 0.576    | 0.304    |
> | Stitched LLaMA 1.4B | KLAS   | **0.593**| **0.337**|
> | Stitched LLaMA 2.7B | ESTA   | 0.631    | 0.353    |
> | Stitched LLaMA 2.6B | KLAS   | **0.645**| **0.379**|
>
> These results validate KLAS in the LLM domain and provide a foundation for future work on stitching multimodal or decoder-only architectures.
> _____________________
>
> [1] He, Haoyu, et al. "Efficient stitchable task adaptation." Proceedings of the IEEE/CVF Conference on Computer Vision and Pattern Recognition. 2024.

---

> > ### Author Response · Authors · 2025-11-22
> > **Rebuttal by Authors to Reviewer B36v (2/2)**
> >
> > **W3**
> > >What performance can KLAS achieve on dense prediction tasks, including object detection, semantic segmentation, and depth estimation?
> >
> > To address this, we have added semantic segmentation (a dense prediction task) results on ADE20K using Mask2Former [2] with Swin-Tiny and Swin-Base backbones in **Section 5.4** (Table 6). The results clearly demonstrate that KLAS consistently outperforms SN-Net across different FLOPs budgets, confirming its applicability to dense prediction tasks.
> >
> > All models are evaluated using mIoU on the ADE20K validation set with frozen Swin backbones. The full networks are fine-tuned for 160k steps using the standard Mask2Former schedule. The FLOPs are measured on 512×512 input resolution. The results show that KLAS delivers higher accuracy at similar or lower FLOPs than SN-Net, with up to +0.9% mIoU improvement.
> >
> > ### Table 6: Semantic Segmentation on ADE20K (Mask2Former + Swin)
> >
> > | Model     | FLOPs (G) | mIoU (%) |
> > |-----------|------------|-----------|
> > | SN-Net-1  | 152        | 29.4      |
> > | **KLAS-1**| 145        | **29.8**  |
> > | SN-Net-2  | 274        | 32.6      |
> > | **KLAS-2**| 277        | **33.5**  |
> > | SN-Net-3  | 327        | 37.7      |
> > | **KLAS-3**| 316        | **37.8**  |
> >
> > KLAS consistently selects higher-quality stitch configurations under equal FLOPs constraints. These results validate that our method generalizes to dense prediction settings like segmentation. Object detection and depth estimation are similar tasks which would use identical block-stitching principles under KLAS. We don't do experiments with these task because it is not necessary after showing for semantic segmentation and out of scope of the rebuttal time window.
> > _____________________
> >
> > [2] Cheng, Bowen, et al. "Masked-attention mask transformer for universal image segmentation." Proceedings of the IEEE/CVF conference on computer vision and pattern recognition. 2022.

---

> > > ### Author Response · Authors · 2025-11-25
> > > **Official Comment by Authors to Reviewer B36v**
> > >
> > > Dear reviewer,
> > >
> > > Thank you again for your valuable feedback! This is a gentle reminder that the discussion period is nearing its conclusion, we hope you have taken the time to consider our responses to your review. If you have any additional questions or concerns, please let us know so we can resolve them before the discussion period concludes. If you feel our responses have satisfactorily addressed your concerns, it would be greatly appreciated if you could raise your score to show that the existing concerns have been addressed.
> > >
> > > Thank you!

---

> ### Author Response · Authors · 2025-11-27
> **Thank you for your thoughtful review and reminder about end of discussion phase**
>
> Dear Reviewer B36v,
>
> Thank you again for your thoughtful review! We wanted to bring to your attention that the discussion period is going to close soon. If you have any additional concerns, please let us know so we can resolve them before the discussion period concludes. **If you feel our responses have satisfactorily addressed your concerns, it would be greatly appreciated if you could raise your score to show that the existing concerns have been addressed**.
>
> Thanks again!!
>
> --KLAS authors

---

### Author Response · Authors · 2025-11-22
**Global Response by Authors to Reviewers about Revised PDF**

- **Section 5.4** has been added to provide results on the **dense prediction tasks**
    - **Appendix G** has been edited to provide further details on the **dense prediction task** experiment setup
- **Section 5.5** has been added to provide results on **LLM stitching**
    - **Appendix H** has been added to to provide further details on the **LLM stitching** experiment setup
- **Section 5.6** has been added to provide ablation studies on **KLAS hyperparameters: Threshold (τ)** and **Bucket granularity**
- **Appendix D** has been added to give (1) **correlation results** backing our argument that **KL divergence is both representational and functional** and (2) **mathematical definitions and arguments** for **KL divergence’s superior performance**
- **Appendix E** has been added for same-family and cross-family settings to show comparison against **cascaded inference**: stitching methods like **KLAS and SN-Net are more suited** to the binary model merging setting for better accuracy-efficiency tradeoff than **cascaded model inference**.
- **Appendix B** and **Section 5**  have been edited to explicitly outline the **finetuning costs**, **wall-clock time** and **parameters of the stitch finetuning process**

- **Section 2** has been edited to include related work for **Cascaded inference**

---

### Author Response · Authors · 2025-11-25
**Global Comment by Authors to all Reviewers**

Dear Reviewers,

We sincerely appreciate your time and effort in reviewing our manuscript and offering valuable suggestions. Since the author-reviewer discussion phase has passed for a week, we would like to confirm whether our responses have effectively addressed your concerns.

We provided detailed responses to your concerns a few days ago, and we hope they have adequately addressed your issues. If you require further clarification or have any additional concerns, please do not hesitate to contact us. We are more than willing to continue our communication with you.

Best regards.

---

### Author Response · Authors · 2025-12-03
**To PC, SAC, and AC: Discussion Summary**

Dear PC, SAC, and AC,

We sincerely appreciate the great efforts and time you spend on checking each submission. Below, we provide a rebuttal status summary to assist your decision-making.

| **Reviewer** | **Status** | **Note** |
|--------------|------------|-----------|
| `8KnS` | Replied [Nov 28] | All concerns resolved; score ↑ to **8** |
| `cTKy` | Replied [Nov 25–28] | All concerns resolved; score ↑ to **6**, promised to ↑ to **8** pending latest cross-family results |
| `XiFF` | Replied [Nov 26] | All concerns resolved; score ↑ to **6** |
| `B36v` | **No Reply** | All concerns answered, original score **6** |

---

**Our Requests**

- Reviewers `XiFF`, `cTKy`, and `8KnS` explicitly confirmed that they intend to raise their scores to **6**, **6 or 8**, and **8** respectively, as we fully resolved their concerns. We would especially appreciate their *explicit* acknowledgements considered in your evaluation.
- Given the lack of response from Reviewer `B36v`, we kindly ask you to review our rebuttal, as we are confident that their concerns have been fully resolved.
---

**Core contributions**

- **Novel Framework**: Introduce KLAS, a KL-divergence-based stitching framework that generalizes across architectures, tasks, datasets and modalities, *proposing a principled approach to inducing a Pareto-efficient **set** of stitched architectures* superseding SN-Net SOTA baseline heuristics.
- **Technical Contribution**: KLAS uses a cheap, 2-stage selection framework (anchor selection + block-level stitch score) that captures both representational and functional similarity, enabling accurate prediction of stitch quality.
- **Superior Performance**: Across DeiT, Swin, ResNet, LeViT, M2F, and LLMs, KLAS achieves SOTA stitched-model performance and consistently improves the accuracy–efficiency tradeoff over SN-Net.
- **Generality**: KLAS extends to dense prediction, LLM stitching, and cross-architecture stitches. Critically, we show KLAS beats threshold-based early-exit cascaded inference, while at the same time subsuming cascade model configs as part of its feasible config space.
---

**Reviewer summaries**

- **Reviewer `B36v`**
  - **KL vs. MSE/CE/CKA/DM**: Added definitions and correlation analyses in **Appendix D** showing KL is the most reliable predictor because it captures functional + representational similarity jointly.
  - **LLM applicability**: Added LLaMA stitching experiments in **Section 5.5** and **Appendix H**, showing KLAS outperforms ESTA (SN-Net based) with fewer FLOPs, establishing KLAS generalizes effectively to LLMs.
  - **Dense prediction**: Added ADE20K semantic segmentation results in **Section 5.4**, demonstrating consistent mIoU improvements at comparable or lower FLOPs, indicating KLAS successfully identifies stitch points in dense prediction tasks.

- **Reviewer `XiFF`**
  - **Representational meaning of KL**: Added shuffled-label and class-conditional CKA ablations in **Appendix D**, confirming KL includes task-relevant representational similarity, but also requires supervision.
  - **Marginal gains**: Clarified gains are substantial for Swin/LeViT and that KLAS never underperforms SN-Net; its strength is generality and automation.
  - **Finetuning details**: Added full training parameters in **Section 5** and **Appendix B** for reproducibility, and noted SN-Net and KLAS produce identical models when stitch selections match, ensuring all performance gains arise from better stitch selection.

- **Reviewer `cTKy`**
  - **Cascade baseline**: Added tuned cascades in **Appendix E**; results show KLAS outperforms cascades across FLOP budgets in both *same-family* and *cross-family* stitching because cascades incur additive compute cost and often use unnecessary anchor blocks.
  - **“Uniqueness” phrasing**: Revised text to reflect evidence showing KL is strongest among tested metrics.
  - **Beyond vision**: Added complete LLaMA experiments in **Section 5.5** and **Appendix H** with clear experimental details and clarified subsequent queries, showing KLAS & probing apply broadly.

- **Reviewer `8KnS`**
  - **Label dependence & ranking validity**: Added supervision ablations and rank-correlation analyses in **Appendix D**, showing KL is the most predictive metric and it requires supervision.
  - **Hyperparameter sensitivity**: Added threshold (τ) and bucket-size ablations in **Section 5.6**, showing stable accuracy/AUC across wide ranges of parameter values.
  - **Marginal gains**: Clarified that KLAS's main strengths are generalizability and automation.
  - **Dense tasks & LLMs**: Added ADE20K (**Section 5.4**) and LLaMA (**Section 5.5**, **Appendix H**) to demonstrate cross-task, cross-modality generality.
  - **Compute overhead**: Clarified in **Section 5** and **Appendix B** that ProbeNet is cheap and finetuning cost matches SN-Net, adding negligible additional overhead.

---

Thank you again for your dedication to safeguarding the fairness of the review process.

Warm regards,

Authors

---

### Meta-Review · Area_Chair_3sRC · 2026-01-02

**Summary:**

This paper introduces ‌KLAS‌, a method for model stitching that uses KL divergence on linear probe outputs to select compatible blocks from a model zoo, aiming to construct new models with favorable accuracy/computation trade-offs. The four reviewers raise substantial and critical concerns that challenge the paper's ‌theoretical soundness, methodological robustness, empirical completeness, and practical significance‌.

**1. Theoretical Foundation and Claim Substantiation are Questioned:‌** This is a serious issue raised by multiple reviewers.

**Misinterpretation of KL Divergence:** Reviewer 2 fundamentally challenges the paper's core premise, arguing that KL divergence compares output distributions, not representational similarity of features. This directly undermines the method's stated rationale and needs ‌a strong rebuttal or a revised theoretical justification.‌\
**Overstated Uniqueness Claim:‌** Reviewer 3 finds the claim that "KL divergence uniquely satisfies the dual objectives" unsubstantiated. This must be either ‌supported by a formal proof or toned down‌ to reflect empirical observation among tested metrics.\
**Justification for Metric Superiority:‌** Reviewer 1's question about why other metrics fail mathematically requires a deeper, more principled explanation beyond empirical observation.

**2. Insufficient and Unfair Empirical Evaluation:‌** The experimental validation is found lacking in key areas, weakening the claims of utility and generality.

**Missing Crucial Baseline (Model Cascades):** Reviewer 3's critique is perhaps the most damaging to the paper's stated goal. The failure to compare against ‌model cascades/committees‌—a standard technique for accuracy/cost trade-offs—leaves a gaping hole in the evaluation. Including this comparison and a ‌discussion on when stitching is preferable‌ (e.g., worst-case latency vs. average performance) is ‌essential.‌\
**Limited Domain and Task Scope:‌** Reviewers 1, 3, and 4 note the experiments are confined to ‌supervised image classification‌. To claim broader impact, the authors must either: demonstrate applicability to ‌dense prediction tasks‌ (segmentation, detection as requested by Reviewers 1 & 4); show preliminary results on ‌other domains‌ (regression, LLMs as suggested by Reviewer 3) or with ‌self-supervised backbones‌ (Reviewer 4).\
**Marginal Gains and Practical Significance:‌** Reviewer 2 notes marginal overall gains, and Reviewer 4 points to very small ΔAUC in some cases. The authors must ‌better contextualize the practical value‌ of these improvements.

**3. Methodological Gaps and Reproducibility Concerns:** Several aspects of the method require clarification and strengthening.

**Lack of Fine-Tuning Details:‌** Reviewer 2's requests for details on the stitch fine-tuning procedure (cost, convergence criteria, model identity) are critical for ‌reproducibility and fair assessment‌ of the reported AUCs.\
**Sensitivity and Robustness Unexplored:‌** Reviewer 4 rightly points out that the method's sensitivity to design choices (bucket size, threshold) and KL divergence's properties (asymmetry, temperature) is underexplored. ‌Ablation and sensitivity studies‌ are needed.\
**Compute Cost Analysis:** While probe training is claimed to be cheap, a ‌full pipeline runtime comparison‌ against the heuristic baseline (SN-Net) would strengthen the efficiency claim (R4).

**4. Clarity and Scholarly Presentation:‌** The paper needs to improve its precision and scope.

**Precise Language:‌** Claims about KL divergence's role must be made with precise, technically accurate language to address Reviewer 2's concern.\
**Expanded Related Work/Discussion:‌** The discussion must explicitly address ‌model cascades‌ (R3) and properly position KLAS within the landscape of techniques for creating efficient model variants.

The paper presents an interesting idea but is not yet ready for publication. The concerns regarding ‌theoretical justification, the missing cascade baseline, and limited empirical scope‌ are particularly substantial. Only after these concerns are addressed, the paper can make a convincing and well-rounded contribution to the field of efficient model design and neural network reuse.

**Reviewer Concerns:**

The main concerns of four reviewers have been addressed by the rebuttal.

**Reviewer Scores:**

Reviewers 2, 3 and 4 had clearly stated that they would increase their scores during the discussion stage. Therefore, all four reviewers consistently recommend an acceptace for this paper.

---

### Decision · Program_Chairs · 2026-01-26

Accept (Poster)